# Numerical simulation and evaluation of global ultrafine particle concentrations at the Earth's surface

Matthias Kohl[1], Jos Lelieveld[1,2], Sourangsu Chowdhury[3], Sebastian Ehrhart[4], Disha Sharma[1], Yafang Cheng[1], Sachchida Nand Tripathi[5], Mathew Sebastian[6], Govindan Pandithurai[7], Hongli Wang[8], and Andrea Pozzer[1,2]

[1]Max Planck Institute for Chemistry, Mainz, Germany
[2]Climate and Atmosphere Research Center, The Cyprus Institute, Nicosia, Cyprus
[3]CICERO Center for International Climate Research, Oslo, Norway
[4]Finnish Environment Institute (SYKE), Marine Research Centre, Helsinki, Finland
[5]Department of Civil Engineering, Indian Institute of Technology Kanpur, Kanpur, India
[6]Centre for Earth, Ocean and Atmospheric Sciences, University of Hyderabad, Hyderabad, India
[7]Indian Institute of Tropical Meteorology, Ministry of Earth Sciences, Pune, India
[8]State Environmental Protection Key Laboratory of Formation and Prevention of Urban Air Pollution Complex, Shanghai Academy of Environmental Sciences, Shanghai 200233, China

**Correspondence:** Matthias Kohl (m.kohl@mpic.de), Andrea Pozzer (andrea.pozzer@mpic.de)

**Abstract.** A new global dataset of annual averaged ultrafine particle (UFP) concentrations at the Earth's surface for the years 2015-2017 has been developed through numerical simulations using the ECHAM/MESSy Atmospheric Chemistry model (EMAC). We present total and size-resolved concentrations along with their interannual variability. Size distributions of emitted particles from the contributing source sectors have been derived based on literature reports. The model results of UFP concentrations are evaluated using particle size distribution and particle number concentration measurements from available datasets and the literature. While we obtain reasonable agreement between the model results and observations (logarithmic scale correlation of $r = 0.76$ for non-remote, polluted regions), the highest values of observed, street-level UFP concentrations are systematically underestimated, whereas in rural environments close to urban areas the model generally overestimates observed UFP concentrations. As the relatively coarse global model does not resolve concentration gradients in urban centres and industrial UFP hotspots, high-resolution data of anthropogenic emissions is used to account for such differences in each model grid box, obtaining UFP concentrations with unprecedented $0.1° \times 0.1°$ horizontal resolution at the Earth's surface. This observation-guided downscaling further improves the agreement with observations, leading to an increase of the logarithmic scale correlation between observed and simulated UFP concentrations to $r = 0.84$ in polluted environments (and 0.95 in all regions), a decrease of the root mean squared logarithmic error (from 0.57 to 0.43), and removes discrepancies associated with air quality and population density gradients within the model grid boxes. Model results are made publicly available for studies on public health and other impacts of atmospheric UFPs, and for intercomparison with other regional and global models and datasets.

# 1 Introduction

Atmospheric aerosols in various size ranges have a significant impact on public health, the hydrological cycle and climate. Close to the Earth's surface, aerosol particles are among the main pollutants and drivers of atmospheric chemistry in the boundary layer, being directly relevant for human health (Burnett et al., 2014; Pope and Dockery, 2006; Cohen et al., 2005). At the same time, aerosols can directly scatter and absorb solar and thermal radiation, altering the radiative balance of the Earth's atmosphere (e. g. Bellouin et al., 2020). Furthermore, atmospheric aerosols act as cloud condensation nuclei (CCN), and thus influence cloud formation processes and cloud properties. Consequently, changes in CCN concentrations may affect the hydrological cycle and indirectly the radiative balance of the Earth's atmosphere by altering cloud cover and albedo (e. g. Christensen et al., 2020; Lohmann and Feichter, 2005; Twomey, 1959).

Recently, multiple studies have concluded that the exposure to particulate matter air pollution from a variety of sources has major implications for public health (Lelieveld et al., 2020, 2015; Chowdhury et al., 2022). The latest Global Burden of Disease study has associated 4.2 million deaths globally to the exposure to ambient particulate matter with an aerodynamic diameter smaller than 2.5 µm ($PM_{2.5}$) and 0.37 million to ambient ozone pollution (Murray et al., 2020). Evidence presented in recent studies indicates that the long-time exposure to high concentrations of ultrafine particles (UFPs), i.e. particles with an aerodynamic diameter smaller than 100 nm (WHO, 2006) which barely contribute to $PM_{2.5}$ mass, significantly impacts human health, leading to the increased incidence of cardiovascular and cerebrovascular diseases (Downward et al., 2018; Delfino et al., 2005; Stone et al., 2017). The health impacts of UFPs may be attributed to their high potential to penetrate more deeply into the lungs and potentially into the blood stream compared to coarser size particles (Schraufnagel et al., 2019; Schraufnagel, 2020; Hong and Jee, 2020).

Reflection and absorption of solar radiation can be neglected for UFPs as the scattering and absorption coefficients typically peak at particle diameters between 400 and 1000 nm. However, aerosol particles with a minimum diameter of approximately 40 nm and larger can serve as CCN, while particle numbers strongly decrease for sizes greater than 200–300 nm in general (Andreae, 2009). Thus, UFPs significantly contribute to CCN concentrations.

Atmospheric aerosols can either be directly emitted into the atmosphere (primary) by natural or anthropogenic processes or nucleate from precursor gases (secondary), with the latter being considered to be the largest source of atmospheric aerosols (Gordon et al., 2017). Freshly nucleated secondary particles usually have a diameter between 1 and 20 nm (Curtius, 2006) and can grow by coagulation and condensation of trace gases (Kulmala et al., 2004). While primary aerosols from natural sources (e. g. desert dust and sea salt) are emitted at diameters predominantly in the micrometer range and larger (Dentener et al., 2006), anthropogenic particles (e. g. from combustion processes) are usually emitted at much smaller sizes, contributing to UFPs (Kwon et al., 2020; Paasonen et al., 2016).

Despite the importance of UFPs for atmospheric processes and human health, very little is known about their global distribution at the Earth's surface. A simulation of global particle number concentrations at the Earth's surface was performed by Gordon et al. (2017), however focusing on new particle formation and CCN up to an altitude of 460 m above the surface, and by Chen et al. (2021) with a nested high-resolution simulation over East Asia ($0.33° \times 0.33°$) and a particular focus on or-

ganic aerosols. Local (surface or vertical) distributions of particle number concentrations and size distributions were measured and modelled by Ketzel et al. (2021) and Frohn et al. (2021, both street- and address-level UFP concentrations in Denmark), Fountoukis et al. (2012) and Saha et al. (2021, high resolution UFP concentrations over Europe and the USA, respectively),

Kukkonen et al. (2016, dispersion of particle numbers in European cities), Franco et al. (2022, NPF and growth in the Amazon rainforest), Williamson et al. (2019) and Liu and Matsui (2022, both new particle formation (NPF) and contribution of organic aerosols in the remote atmosphere from the surface to the upper troposphere), and Weigel et al. (2021, NPF in the South Asian monsoon).

Thus, present knowledge on global surface UFP concentrations is mostly limited to local model studies and in-situ mea-

surements of particle size distributions (PSDs), from which UFP concentrations can be inferred. While the number of PSD measurements is increasing (Wu and Boor, 2021; Rose et al., 2021), they are still sparse and mostly not continuous over time. Furthermore, there are no clear methodological guidelines for measuring PSDs or particle number concentrations (PNCs) (Trechera et al., 2023) whereas new recommendations have been recently suggested (CEN/TS 17434:2020, 2020; CEN/TS 16976:2016, 2016; ACTRIS, 2021), and measurement size limits vary greatly.

To generate a first, global, annually averaged UFP dataset for the years 2015-2017 we used the ECHAM/MESSy Atmospheric Chemistry model (EMAC, Jöckel et al., 2006), including gas phase and heterogeneous chemistry with comprehensive chemical mechanisms, aerosol microphysics with size-resolved particulate matter and cloud interactions. Such data can be applied to investigate the impact of UFPs on public health, CCN formation and cloud properties, as well as for intercomparison studies with other regional and global models and datasets.

Several emission inventories describe the total emitted mass of aerosol and gas species on a global grid (e. g. Hoesly et al., 2018; Crippa et al., 2020; Granier et al., 2019). However, the inference of the number of emitted particles is very sensitive to their size distribution as small changes can lead to large deviations in the resulting particle number concentrations. Information on these size distributions is rare and the uncertainties are high (Paasonen et al., 2016). For that reason we use existing information on PSDs of emitted particles from the literature, and evaluate our results against measurement data in

China, India, Europe, the United States and various remote regions.

Another challenge in the global modelling of UFPs is the limited model resolution. Studies showed that UFP concentrations return to background levels within a distance of about $1000\,\mathrm{m}$ from the source (e. g. Karner et al., 2010, for roadways). Thus, UFP concentrations can show sharp urban to rural gradients (e. g. Saha et al., 2021) that cannot be captured efficiently by a global model that is limited in the horizontal grid size. As a result, high local UFP concentrations in densely populated

urban areas may be artificially diluted by the surrounding regions in the grid boxes. This may result in an underestimation of UFP concentrations in city centres, while concentrations in remote regions close to urban areas may be overestimated. This shortcoming in the evaluation of model results is studied in depth as the inability of localised measurements in representing the grid box environment (representation error) in Schutgens et al. (2016a, b, 2017).

The correlation between the underestimation of the model on the one hand, and local high resolution anthropogenic emis-

sions at the measurement sites relative to the average anthropogenic emissions at model resolution on the other hand is used here to derive UFP concentrations at $0.1° \times 0.1°$ horizontal resolution. We show that this improves the agreement between ob-

servations and simulations, reduces the spatial representation error, and decreases inconsistencies introduced by the difficulties of the coarse model resolution to capture population density and air pollution gradients.

## 2 Global Model & Methods

The ECHAM/MESSy Atmospheric Chemistry (EMAC) model (Jöckel et al., 2006) is a combination of the 5th generation European Centre Hamburg general circulation model, ECHAM5, (Roeckner et al., 2003), which serves as the dynamical basemodel, and the second version of the Modular Earth Submodel System, MESSy2, (Jöckel et al., 2010), comprising various submodels that describe the chemistry and physics of the atmosphere.

The simulation used for this study is performed at a spectral, horizontal resolution of T63 ($1.875° \times 1.875°$ or approx. $180 \times 180\,\mathrm{km}$ at the equator) with 31 vertical, hybrid terrain-following and pressure levels up to $10\,\mathrm{hPa}$ altitude, and the surface level extending up to approximately 45-70 m above the surface, depending on latitude and season. The model simulation is "nudged" (Jeuken et al., 1996; Jöckel et al., 2006) towards meteorological reanalysis data of the years 2014-2017 (ERA-Interim, Berrisford et al., 2011) from the European Centre for Medium-Range weather forecasts (ECMWF).

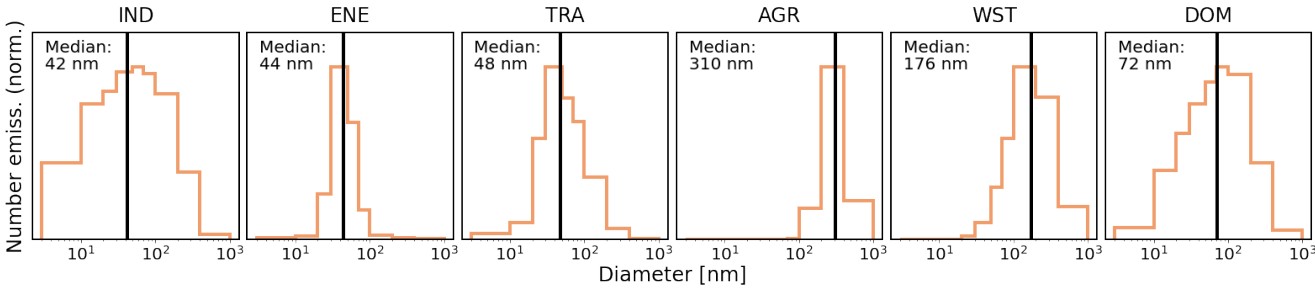

**Figure 1.** Normalized size distributions of primary emitted particles from different sectors (IND: industries, ENE: energy generation, TRA: land transportation, AGR: agricultural soils, WST: waste, DOM: domestic energy use) taken from Paasonen et al. (2016) with corresponding median diameters derived from these distributions (black line). The median diameters are taken as median diameters for the sector emissions in our simulation.

Global anthropogenic emissions of reactive gases and aerosols at the surface from the simulated years are taken from the Community Emission Data System (CEDS McDuffie et al., 2020). We consider primary emitted Black Carbon (BC), Organic Carbon (OC) and sulfate ($SO_4$; 2.5% of $SO_2$ emissions according to Dentener et al., 2006) as direct aerosol sources at the surface. The CEDS anthropogenic emissions from the sectors energy generation (ENE), industries (IND), land transportation (TRA), domestic energy use (DOM), waste (WST), agricultural soils (AGR), solvent production and application (SLV; no primary particle emissions), and ship and other navigation (SHIP) are considered. Emissions from biomass burning (BB) and agricultural waste and residue burning (AWB) were calculated daily using the BIOBURN submodel based on observed dry matter burned and fire type compiled by Andreae (2019). The biomass burning emissions for OC and BC were increased by a factor of 4.48 and 2.8, respectively, based on the work of Pan et al. (2020) and a comparison with observations in the

Amazon Basin (Holanda Bruna, personal communication). Aircraft emissions of reactive gases, BC and OC were taken from the CAMS Global aviation emissions (CAMS-GLOB-AIR; Granier et al., 2019). Seasalt (algorithm from Guelle et al., 2001) and dust emissions (algorithm from Klingmüller et al., 2018) are calculated online using the submodel ONEMIS (Kerkweg et al., 2006). All emissions are distributed in six different emission height levels based on the description by Pozzer et al. (2009).

Aerosols are treated using the MESSy submodel GMXe (Pringle et al., 2010). Aerosol microphysics are based on aerosol size distributions with currently 7 interactive lognormal modes that cover the typical size spectrum of aerosol species and differentiate into 4 hydrophilic (Nucleation, Aitken, Accumulation and Coarse) and 3 hydrophobic (Aitken, Accumulation and Coarse) aerosol modes. All aerosols are approximated as spherical particles. The properties of aerosols in each mode are completely defined by the total mass (internal mixture of contributing species), density, number concentration, median radius and width of the log-normal distribution. After each simulation step aerosols may transfer between modes depending on size changes. Organic aerosol species are additionally described by the ORACLE (Organic Aerosol Composition and Evolution; Tsimpidi et al., 2018) submodel, taking into account the partitioning between aerosols and the gas phase. ORACLE distinguishes between primary and secondary organic aerosols from different sources and volatilities (in up to 5 logarithmically spaced saturation concentration bins, ranging from $10^{-2}$ to $10^6$ ug/m$^3$, depending on the emission sector).

Heterogeneous and gas phase chemistry are treated with the submodel MECCA (Sander et al., 2019) with the Mainz Isoprene Mechanism (MIM1; Pöschl et al., 2000; Jöckel et al., 2006) as chemical mechanism, comprising more than 100 gas phase species and more than 250 reactions. Dry deposition, sedimentation and wet deposition are simulated with the submodels DDEP, SEDI (both Kerkweg et al., 2006) and SCAV (Tost et al., 2006), respectively. The submodel NAN (Ehrhart et al., 2018) is used to estimate binary and ternary nucleation following Dunne et al. (2016), pure organic nucleation (Kirkby et al., 2016) and nucleation from oxidized organic species and sulfuric acid (Riccobono et al., 2014). The parameterization of ion induced nucleation is included in NAN, using ion pair production and steady-state ion concentrations from the submodel IONS (Ehrhart et al., 2018). IONS calculates the ion pair production from galactic cosmic rays and from radon decay, the latter obtained from the diagnostic radon (DRADON) submodel (Jöckel et al., 2010).

Aerosols at the surface simulated with the EMAC model have been extensively evaluated in many publications, either with a focus on $PM_{2.5}$ mass or on aerosol optical depth (Pozzer et al., 2012, 2015; Lelieveld et al., 2019; Pozzer et al., 2022). Chowdhury et al. (2022) conducted an evaluation for aerosol optical depth, $PM_{2.5}$, and black carbon and organic aerosols in $PM_{2.5}$ using a similar setup as used in this study.

The number of emitted aerosols ($N_{aer}$) is calculated as

$$N_{aer} = \frac{6 \cdot M_{aer}}{\pi \cdot \rho_{aer} \cdot d_{med}^3} \cdot \exp\left(-4.5 \ln^2 \sigma_{ln}\right) \tag{1}$$

where the emitted aerosol mass $M_{aer}$ is given by the respective emission dataset. $\rho_{aer}$ is the density of the considered aerosol species and $\sigma_{ln}$ is the width of the log-normal mode in the model. The fraction is just the geometrical derivation of the number of spherical particles from total mass, given a diameter $d_{med}$, while the exponential function corrects for the lognormal volume distribution (compare equations (8.34) and (8.51) from Seinfeld and Pandis, 2016). The median diameter of the emitted particles

$(d_{med})$ depends on the considered sector and species. As $N_{aer} \propto 1/d_{med}^3$, the number of emitted particles is highly sensitive to the emission median diameter, i. e. a doubling of the diameter leads to a reduction of the number of emitted particles by a factor of 8.

In order to emit the aerosols with a realistic size distribution for each sector, a detailed investigation of the emission size distributions is performed based on the findings of Paasonen et al. (2016). Paasonen et al. (2016) used an emission model (Amann et al., 2011) in combination with emission factors and size distributions from the literature in order to obtain global emission particle size distributions for different sectors. As the median emission diameter $d_{med}$ is a global quantity in the EMAC model, we derived the median of the present global size distributions from Paasonen et al. (2016) and used them as

$d_{med}$ for the corresponding sectors in our model. The distributions along with their median diameters are depicted in Fig. 1.

        Additionally, the aerosol median emission diameter from biomass burning and AWB is estimated to be 130 nm based on the average of multiple studies on biomass burning emissions summarized in a review (Reid et al., 2005), confirmed by Janhäll et al. (2010, who measured 120 nm) and the respective size distribution from Paasonen et al. (2016) for AWB (median diameter of 126 nm). Ship aerosol emissions are assumed to be represented by a median diameter of 40 nm based on studies by Kasper

et al. (2007, who found 20–40 nm for low-speed marine diesel engines) , Diesch et al. (2013, small nucleation mode from 10–20 nm and Aitken mode around 35 nm) and Petzold et al. (2008, 52 nm in fresh plume up to 100 nm in aged plume). Aerosols from aircraft emissions are emitted at a diameter of 40 nm as well, based on studies from Petzold and Schröder (1998) and Petzold et al. (2003), who distinguished a mode between 30 and 45 nm and an additional smaller accumulation mode around 180 nm. Dust particles are emitted only in the accumulation and coarse modes according to Klingmüller et al. (2018) and play a

negligible role in UFP concentrations (d'Almeida, 1987). For the emissions of sea salt we use the diameters from the algorithm of Monahan (1986), also only emitting particles in the accumulation and coarse modes.

        As only few measurements of PSDs are available, which can be used to directly infer UFP concentrations, measurements of total PNCs are additionally used for the model evaluation as they are often dominated by UFPs, especially close to UFP sources (e. g. Baldauf et al., 2016; Kumar et al., 2014). Figure 2 shows a typical simulated PSD (blue line) as the sum of the

four soluble and three insoluble modes. The red solid line shows the upper bound for UFPs at 100 nm. The red dashed line is the variable lower bound, used exclusively for comparison to observations, as measurement devices entail a cutoff particle diameter below which no particles are detected. The published global UFP dataset includes arbitrarily small particles and does not use a lower cutoff value.

        The UFP concentration in the simulation $N_{\mathrm{UFP}}$ is calculated using the particle number concentration $N_{tot,i}$, the width $\sigma_i$

and the median diameter $D_{m,i}$ of each mode $i$:

$$N_{\mathrm{UFP}} = \sum_{i=1}^{7} \frac{N_{tot,i}}{2} \cdot \left\{ \mathrm{erf}\left( \frac{\ln\left(D_{up}/D_{m,i}\right)}{\sqrt{2}\ln\left(\sigma_i\right)} \right) \right.$$
$$\left. - \mathrm{erf}\left( \frac{\ln\left(D_{low}/D_{m,i}\right)}{\sqrt{2}\ln\left(\sigma_i\right)} \right) \right\} \tag{2}$$

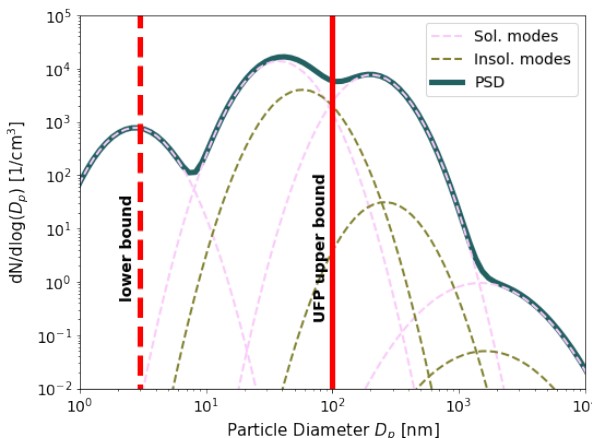

**Figure 2.** Typical particle size distribution (PSD) taken from the simulation in an urban region. The dashed lines represent the soluble and insoluble aerosol modes. The PSD is the sum of all these modes (blue line), typically dominated by the soluble modes. UFPs are defined as all particles with diameter below 100 nm (right red line), while the total particle number concentration is the full integral over the PSD. For comparison between observed and simulated concentrations, the lower bound (red dashed line) is considered as a cutoff, which depends on the detection limit of the measurement device. The final dataset includes all UFPs without lower bound.

where $D_{up}$ is the fixed upper bound of $100$ nm and $D_{low}$ is the variable lower detection limit associated with the measurement device below which no particles can be detected. The cut on the detection limit is applied on the simulation for comparisons with observations. For the final global dataset we report the total number of UFPs, and thus $D_{low}$ is set to 0 nm

and the second error function in Eq. (2) evaluates to $-1$.

The total PNC is the sum over the particle number concentrations $N_{tot,i}$ for each mode (i. e. $D_{up} = +\infty$ and the terms in the curly brackets in Eq. (2) add up to $+2$ for $D_{low} = 0$). However, PNC measurements also have a lower cutoff value, which has to be considered for model evaluation when comparing simulated and measured PNCs.

## 3   Observations

In order to evaluate the simulated UFP concentrations from the EMAC model, observations were collected from different stationary measurement sites at the Earth's surface with a focus on polluted regions. The simulation has been performed for the years 2015-2017 with one year of spin-up (2014), whereas we evaluated the simulation for the year 2015 only. Thus, if available, the simulations were compared to observations in 2015. However, to increase the number of available datasets to compare with simulation results, we additionally used annual averages of observed UFP concentrations and PNCs from all

available years for the evaluation. This is noted accordingly in the following sections. Observational data were collected for UFP concentrations and PNCs and compared to the respective calculated simulated values according to the lower cutoff value

of the respective measurement devices. The sources of the observational data of UFPs (derived from PSDs) and PNCs are listed below.

## 3.1 EBAS

EBAS is a database for atmospheric measurement data operated by the Norwegian Institute for Air Research (NILU) and contains measurements for different programs of which we used the following:

- **EMEP** (European Monitoring and Evaluation Programme) monitors air pollutants in Europe.

- **GAW-WDCA** (Global Atmosphere Watch - World Data Centre for Aerosols) is a data repository for microphysical, optical and chemical properties of atmospheric aerosol.

- **ACTRIS** (Aerosol, Cloud and Trace Gas Research Infrastructure) contains long-term atmospheric measurement data.

The data was obtained from the EBAS database (http://ebas.nilu.no/, last access on February 17, 2022). We analysed all available PSDs (mostly Europe and remote regions) and PNC measurements (Europe, Northern America and remote regions) taken in 2015.

## 3.2 Field measurements, literature and published datasets

We derived UFP concentrations from PSDs measured by groups involved in the present study in India, China and the Amazon rainforest. In India PSDs were measured in Delhi (Thamban et al., 2021), Mahabaleshwar and Hyderabad (both Sebastian et al., 2022). Measurements in China were taken in Shanghai (unpublished), Beijing (Liu et al., 2020), Lin'an (Shen et al., 2022) and Gucheng (Li et al., 2021). Additionally, we used measurements from the Amazonian Tall Tower Observatory (ATTO) centrally located in the rainforest in Brazil, about 150 $\mathrm{km}$ northeast of Manaus (Franco et al., 2022). The observational datasets are 205 complemented by including literature and published datasets from China and India (Gani et al., 2020; Wu et al., 2008), as well as from the ATom aircraft campaign (Brock et al., 2019).

## 4 Results

Annual averages of modelled UFP (UFP$_\mathrm{M}$) concentrations in 2015 on a global scale are displayed in Fig. 3, ranging from about $40$ per $\mathrm{cm}^3$ over parts of the ocean up to more than $12,000$ per $\mathrm{cm}^3$ in China, India, Indonesia and Papua New Guinea 210 (considering averages over model grid boxes). UFP$_\mathrm{M}$ concentrations are generally high across Europe ranging from $2,000$ to $6,000$ per $\mathrm{cm}^3$, however, lower than in several Asian hotspot areas. In North America, UFP$_\mathrm{M}$ concentrations are simulated to be higher along the East coast, with additional hotspots in and around the big cities along the West Coast, reaching up to $8,000$ per $\mathrm{cm}^3$ in and around San Francisco and Los Angeles. We also simulate high UFP$_\mathrm{M}$ concentrations close to $10,000$ per $\mathrm{cm}^3$ at the West Coast of South America and regional enhancements along the East Coast. Southern Australia, Northern Africa, the 215 Middle East (the Persian Gulf region) also show enhanced UFP$_\mathrm{M}$ concentrations exceeding $8,000$ per $\mathrm{cm}^3$.

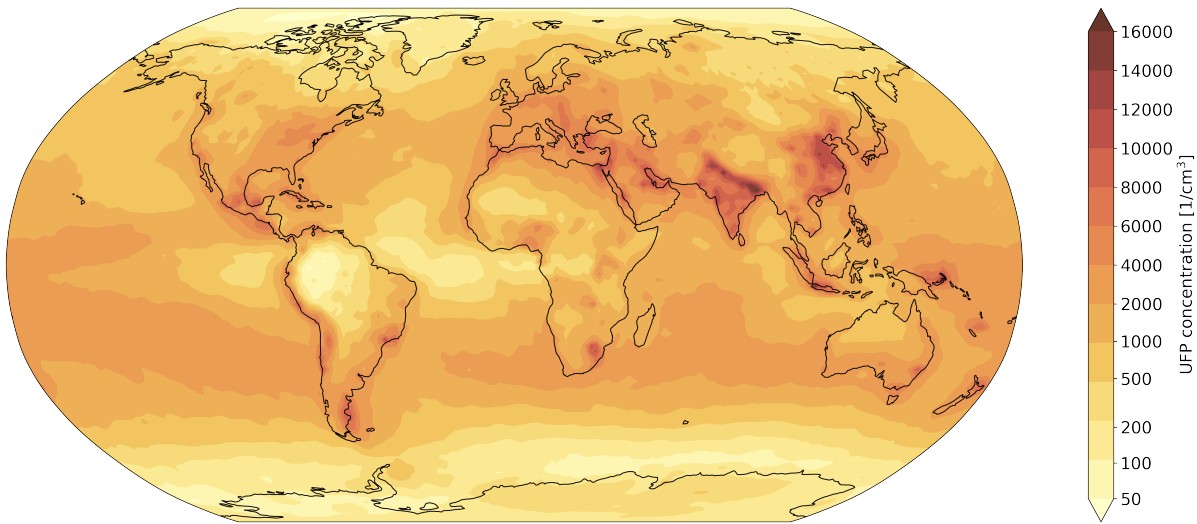

**Figure 3.** Annual average of global UFP concentrations simulated with EMAC for the year 2015 at model resolution of $1.875° \times 1.875°$ and from the lowest vertical model level (surface level).

Continental polar regions in North America, Europe, Asia and Antarctica are simulated to have $UFP_M$ concentrations mostly below 500 per $cm^3$, due to the absence of pollution particle sources. Low $UFP_M$ concentrations (down to 150 per $cm^3$) are also simulated across the Amazon rainforest and in major desert regions (e.g. the Sahara). However, there is a lack of measurement data of PNCs and UFP concentrations in desert regions to evaluate the model results. Low UFP concentrations in the boundary
layer over the Amazon forest have been reported previously and were attributed to missing sources of primary UFPs and the growth of secondary particles during transport from the upper troposphere through the condensation of oxidized organic species, reaching the boundary layer through convective downdrafts (Zhao et al., 2020; Wang et al., 2016; Andreae et al., 2018).

$UFP_M$ concentrations over the oceans are highly variable, with relatively high values exceeding $2,500$ per $cm^3$ over the Pacific and Indian Oceans downwind of pollution sources on land, and very low values below 50 per $cm^3$ over the Southern
Ocean and tropical Atlantic and Pacific Oceans. Low UFP concentrations in the tropical ocean environment have been observed by the ATom aircraft campaign as well, potentially caused by the efficient removal of small particles by coagulation, and again the downward transport of aged secondary particles from the upper troposphere (Williamson et al., 2019).

PNCs were modelled previously by Gordon et al. (2017) at relatively low global resolution, by Saha et al. (2021) for the United States only, and by Chen et al. (2021) with a focus on East Asia. Gordon et al. (2017) concentrated on cloud
condensation nuclei and averaged over a vertical column of 460 m. We generally simulate higher concentrations compared to Gordon et al. (2017) mostly capturing UFP hotspots, and in accord with observations, especially after the redistribution based on anthropogenic emissions (see Sect. 4 and 4.2). The comparably lower concentrations simulated by Gordon et al. (2017) might partially result from the larger vertical column in the simulation, the coarser resolution or the different emission diameters (globally 60 nm for all fossil fuel related emissions). Our simulated UFP concentrations agree very well with the

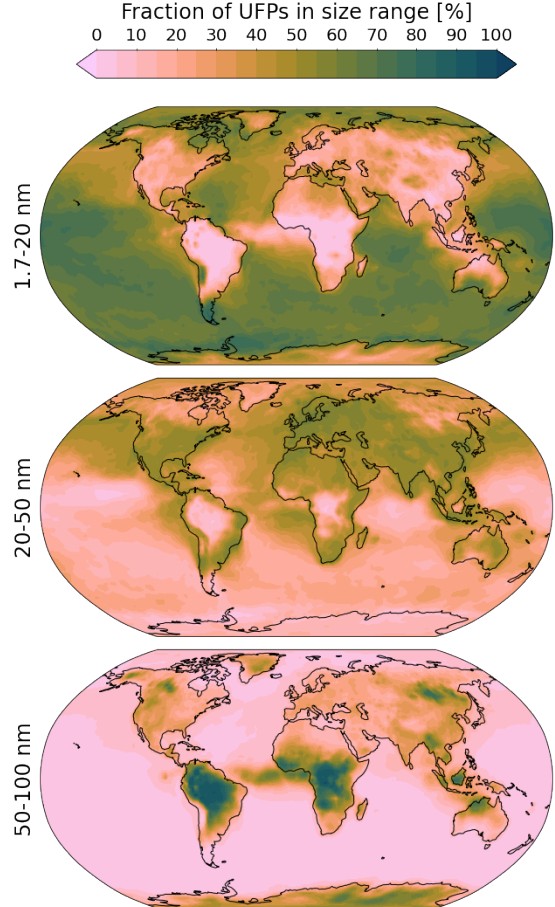

**Figure 4.** Fraction of UFPs in different size ranges, representing freshly nucleated particles (1.7 - 20 nm), fossil fuel emissions or grown nucleation particles (20 - 50 nm), and further grown or larger emitted particles (50 - 100 nm).

PNC simulation from Chen et al. (2021) in most continental regions, especially Europe, North and South America, Africa, Australia and East Asia (which they focused on). However, simulated PNCs in India are considerably lower than our UFP concentrations (supported by observations, see Sect. 4.1.3). Additionally, we simulate higher UFP concentrations at marine southern mid-latitudes (in reasonable agreement with measurements during the ATom campaign, see Sect. 4.1.5) than Chen et al. (2021).

The simulated spatial distribution of UFPs over contiguous USA matches that of Saha et al. (2021). They used a land use regression model to produce a high (200 × 200 m) resolution product of UFP concentrations. We reach similar values in the urban regions in the USA after the redistribution based on anthropogenic emissions as described in Sect. 4.2, however still lower than the highest values simulated by Saha et al. (2021). The difference might be due to the fact that UFP concentrations are a subset of PNCs. Ketzel et al. (2021) simulated PNCs at street- and address-level in the same order of magnitude

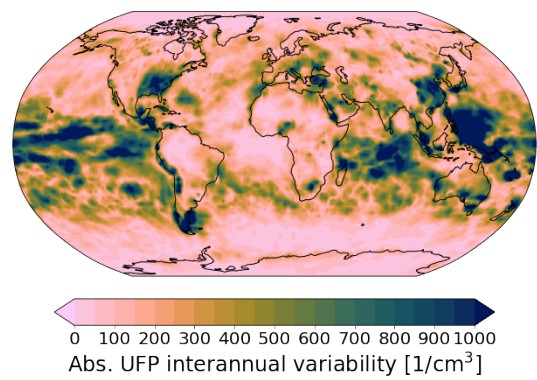

**Figure 5.** Absolute interannual variation of UFP concentrations calculated based on the years 2015-2017. It is defined by the maximum minus the minimum annual averaged UFP concentrations.

as our simulation up to urban background level, but with much higher peak values reaching more than $30,000$ per $\mathrm{cm}^3$ in traffic hotspots, which we cannot resolve. Our simulated UFP concentrations are much lower than the modelled results from Fountoukis et al. (2012), that reach around $20,000$ per $\mathrm{cm}^3$, even for background regions in Europe in May 2008, and up to more than $100,000$ per $\mathrm{cm}^3$ in local hotspots in Southeastern Europe. However, the general increasing trends from Western to Eastern Europe agree.

Trechera et al. (2023) analysed observations of PNCs and UFP concentrations in Europe from 2017 to 2019, focusing on daily and seasonal patterns, UFP drivers and regional trends. They again find increasing UFP concentrations from Northern to Southern and from Western to Eastern Europe. Our simulation also exhibits the west-east tendency in Europe, however no clear pattern from North to South. Seasonality and drivers of UFP concentrations and composition will be the subject of follow-up studies.

UFPs have widely differing size distributions across the globe. Figure 4 illustrates this based on three size bins, i. e. 1.7-20 nm, 20-50 nm, and 50-100 nm. Freshly nucleated particles between 1.7 and 20 nm mostly dominate in remote environments, especially over oceans. PSDs usually peak at diameters between 20 and 50 nm in polluted regions, representing a mixture of freshly emitted fossil fuel primary and rapidly grown secondary particles. Biomass burning particles are typically emitted at larger diameters (see Fig. 1). Additionally, high concentrations of organic vapors lead to strong condensational growth in forest regions. Thus, UFPs are dominated by 50-100 nm particles there.

The interannual variability in UFP concentrations is presented in Fig. 5. We observe strong interannual variation, exceeding 1000 per $\mathrm{cm}^3$ and (up to 50 %) along the ITCZ, that is mostly caused by meteorological differences in the different years, potentially caused by the strong El Niño in 2015, followed by weaker La Niña and El Niño events in 2016 and 2017. The meteorological influence on UFP concentrations could be the basis for future studies. We additionally find absolute interannual variation exceeding 1000 per $\mathrm{cm}^3$ over polluted regions, although below 20 % in relative terms. This is mostly due to a decreasing trend in emissions over South and East Asia, Europe and the United States. However, we observe an increasing trend in UFP concentrations over parts of Africa, especially Nigeria and South Africa.

### 4.1 Evaluation

In the following, we present the evaluation of the simulated $UFP_M$ concentrations based on observational data. Sections 4.1.1 to 4.1.4 evaluate urban and rural regions in four comparably highly polluted regions for which measurement data are available, namely Europe, North America, India and China. In Sect. 4.1.5 remote regions in polar, forest, mountain and ocean environment with lower population density are evaluated. Mountain environments are defined covering altitudes from 1500 m to 3000 m asl, while measurements above 3000 m asl were excluded as they are mostly located in the free troposphere, and this study concentrates on boundary layer processes and the respective evaluation.

Annual averages of observed UFP ($UFP_O$) concentrations, respectively PNCs ($PNC_O$), were either derived from daily PSDs (by integrating the number of aerosol particles per size bin from the lowest measurement bin up to the bin crossing the upper size threshold of 100 nm) and PNCs – removing outliers differing by a factor greater than 10 from the median value, and subsequently calculating the annual average –, or directly provided. The daily averaged number concentrations of the model aerosol modes were integrated for the same size region as the observations (from the lower detection limit of the measurement device up to the highest measurement bin with a mean diameter below 100 nm) according to Eq. (2). $UFP_M$ concentrations were sampled at the vertical grid box covering the measurement site altitude.

The evaluation in this section concentrates on the modelled ($UFP_M$ or $PNC_M$) and observed ($UFP_O$ or $PNC_O$) concentrations. The downscaled (i. e redistributed within each grid box) concentrations ($UFP_R$ or $PNC_R$) will be the subject of Sect. 4.2. We used three different statistical measures for the evaluation based on daily model output and observations if available:

– PF2: Percentage of modelled values that are simulated within the range of a factor of two of the observed values

– NRMSE: RMSE (root mean squared error) normalized by the range of the observations

– $\overline{M/O}$: Geometric mean of the ratio between modelled (M) and observed daily mean (O)

#### 4.1.1 Europe

$UFP_O$ concentrations in Europe were derived from daily PSDs provided by the EBAS database. An overview of the different measurements in Europe and a comparison to $UFP_M$ is presented in Table 1. We used all available observations from the EBAS database from 2015 in Europe from rural and urban stations. The remote stations as defined above were excluded here and will be separately discussed in Sect. 4.1.5. Additionally, we excluded measurements in Athens (GR) and Preila (LT) due to apparent inconsistencies in the observational datasets.

The simulation shows reasonable agreement with the observations at most measurement stations with $PF2 \geq 40\%$, $NRMSE \leq 0.4$ and $0.5 \leq \overline{M/O} \leq 2.0$. However, there are some exceptions. At the measurement stations of Madrid, Leipzig-Eisenbahnstrasse, Leipzig-Mitte and Dresden-Nord the model strongly underestimates the UFP concentrations. This is due to a combination of two effects. Firstly, UFP concentrations are typically related to anthropogenic emissions (see Sect. 4.2 for details). The high UFP concentrations in the densely populated urban centres are artificially diluted in the simulation by lower concentrations in the surroundings covered by the grid box. This spatial representation error after Schutgens et al. (2016a) can partly be

**Table 1.** Summary of observations of UFP (UFP$_O$) concentrations ($\mathrm{cm^{-3}}$) derived from PSDs across Europe for 2015, with data archived in the EBAS database. The geographical location is indicated by latitude (Lat) and longitude (Lon), the elevation of the site above the sea surface is indicated by the altitude asl in metres (Alt). The lower cutoff value (Cut) from the measurement device (also applied on the simulation) is expressed in nm. The annual averages of the UFP$_O$ and UFP$_M$ concentrations for the grid box encompassing the station are compared and different measures of agreement are listed. The same comparisons are performed after the redistribution of the model ("Redistributed model", UFP$_R$), which will be discussed in detail in Sect. 4.2. All included measurement stations contained at least 200 days of valid measurements in 2015. Abbreviations: DD – Dresden, NOAK – National Atmospheric Observatory Košetice, ECO – Environmental-Climate Observatory, LE – Leipzig, OPE – Observatoire pérenne de l'environnement.

| Station | Observations | | | | | Model results | | | | Redistributed Model | | | |
|---|---|---|---|---|---|---|---|---|---|---|---|---|---|
| | Lat | Lon | Alt | Cut | **UFP$_O$** | **UFP$_M$** | PF2 | NRMSE | $\overline{\mathrm{M/O}}$ | **UFP$_R$** | PF2 | NRMSE | $\overline{\mathrm{M/O}}$ |
| Annaberg-Buchholz (DE) | 50.57 | 13.0 | 545 | 9.4 | 6649 | 3597 | 55 | 0.21 | 0.55 | 3317 | 49 | 0.22 | 0.51 |
| Cabauw Zijdeweg (NL) | 51.97 | 4.93 | 1 | 9.1 | 6324 | 4539 | 77 | 0.22 | 0.72 | 3828 | 62 | 0.26 | 0.6 |
| DD-Nord (DE) | 51.06 | 13.74 | 120 | 4.8 | 10526 | 4289 | 29 | 0.34 | 0.39 | 7690 | 67 | 0.25 | 0.7 |
| DD-Winckelmannstr (DE) | 51.04 | 13.73 | 112 | 9.4 | 5474 | 4218 | 85 | 0.16 | 0.78 | 7562 | 77 | 0.22 | 1.39 |
| ECO Lecce (IT) | 40.34 | 18.12 | 36 | 9.4 | 5372 | 3741 | 59 | 0.36 | 0.62 | 4821 | 62 | 0.37 | 0.8 |
| Finokalia (GR) | 35.34 | 25.67 | 250 | 8.6 | 1421 | 3628 | 34 | 0.6 | 2.3 | 3917 | 28 | 0.67 | 2.48 |
| Hohenpeissenberg (DE) | 47.8 | 11.01 | 985 | 9.4 | 2302 | 2479 | 78 | 0.25 | 1.08 | 2234 | 76 | 0.23 | 0.97 |
| K-puszta (HU) | 46.58 | 19.35 | 125 | 6.0 | 4040 | 5127 | 67 | 0.27 | 1.32 | 4966 | 66 | 0.26 | 1.28 |
| LE-Eisenbahnstr (DE) | 51.35 | 12.41 | 120 | 4.8 | 14410 | 4358 | 15 | 0.37 | 0.3 | 5020 | 23 | 0.35 | 0.34 |
| LE-Mitte (DE) | 51.34 | 12.38 | 111 | 4.8 | 10624 | 4251 | 27 | 0.34 | 0.39 | 5521 | 47 | 0.3 | 0.5 |
| LE-West (DE) | 51.32 | 12.3 | 122 | 4.8 | 6222 | 4451 | 74 | 0.21 | 0.69 | 4686 | 77 | 0.21 | 0.73 |
| Leipzig (DE) | 51.35 | 12.43 | 118 | 4.8 | 4882 | 4300 | 88 | 0.17 | 0.86 | 4953 | 91 | 0.17 | 0.99 |
| Madrid (ES) | 40.46 | -3.73 | 669 | 14.4 | 11311 | 2916 | 22 | 0.17 | 0.28 | 6192 | 59 | 0.15 | 0.59 |
| Melpitz (DE) | 51.53 | 12.93 | 87 | 4.8 | 7018 | 4343 | 72 | 0.19 | 0.62 | 4400 | 72 | 0.19 | 0.63 |
| Montseny (ES) | 41.78 | 2.36 | 700 | 8.9 | 3056 | 2375 | 69 | 0.21 | 0.78 | 2371 | 69 | 0.21 | 0.78 |
| NOAK Kosetice (CZ) | 49.58 | 15.08 | 534 | 8.6 | 2569 | 4332 | 57 | 0.3 | 1.79 | 3701 | 66 | 0.24 | 1.53 |
| Neuglobsow (DE) | 53.14 | 13.03 | 62 | 9.4 | 2808 | 3515 | 69 | 0.17 | 1.3 | 2986 | 74 | 0.15 | 1.11 |
| OPE (FR) | 48.56 | 5.5 | 392 | 9.7 | 1834 | 2865 | 63 | 0.32 | 1.6 | 2786 | 64 | 0.31 | 1.55 |
| Prague-Suchdol (CZ) | 50.12 | 14.38 | 277 | 5.6 | 6421 | 4412 | 74 | 0.15 | 0.7 | 6124 | 89 | 0.13 | 0.97 |
| SIRTA Palaiseau (FR) | 48.71 | 2.16 | 162 | 10.0 | 4239 | 3441 | 82 | 0.2 | 0.84 | 5391 | 79 | 0.24 | 1.31 |
| Schauinsland (DE) | 47.9 | 7.92 | 1205 | 9.4 | 1545 | 1833 | 78 | 0.19 | 1.22 | 1783 | 80 | 0.19 | 1.19 |
| Vielsalm (BE) | 50.3 | 6.0 | 496 | 8.8 | 2082 | 2596 | 75 | 0.22 | 1.26 | 2628 | 73 | 0.23 | 1.27 |

corrected for by redistributing UFP concentrations using high resolution anthropogenic emission datasets, discussed in detail in Sect. 4.2. Secondly, the measurements were performed close to busy streets (Leipzig-Eisenbahnstrasse, Leipzig-Mitte and

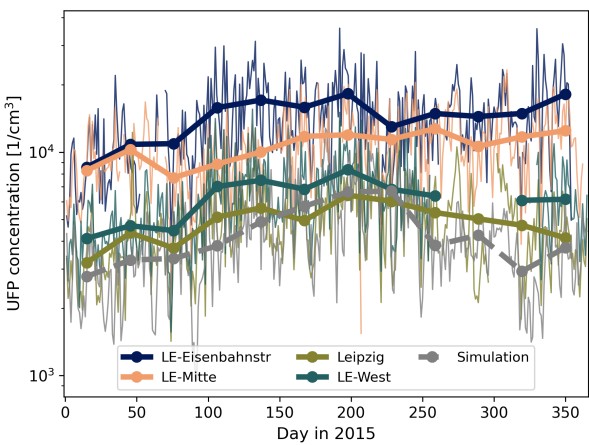

**Figure 6.** Monthly averaged UFP concentrations in the simulation (grey, dashed) and measured at different stations in Leipzig, Germany (thick solid lines). The daily fluctuating UFP concentrations are shown with thin, transparent lines in the same color.

Dresden-Nord sites are explicitly measuring traffic emissions) with high primary UFP emissions from vehicles, which would require impracticable street-level horizontal resolution to resolve.

The model overestimates $UFP_M$ concentrations at the stations of L'Observatoire pérenne de l'environnement (OPE, France) and Finokalia (GR). OPE is located at a remote region close to Nancy and the grid box is thereby highly influenced by the urban region. Finokalia on the other hand is located in a rural environment close to the coast of the island Crete. The interference of the oceanic influence and surrounding enhanced anthropogenic emissions from the island leads to enhanced UFP concentrations within the grid box.

Multiple measurements in one city provide an interesting case of the variability of UFPs within one modelled grid box. Measurements were performed at four different locations in Leipzig (shown in Fig. 6), varying by up to a factor of 3, while simulated $UFP_M$ concentrations (grey dashed line) approximately coincide with the observations taken at some distance from heavy traffic locations (Leipzig). The station at Leipzig-Eisenbahnstrasse is at a curbside, and measures very high $UFP_O$ concentrations. It can be noted that the model is capable of reproducing urban background conditions, i.e. the average over urban regions and their surroundings in the grid box, while not replicating local UFP hotspots, for example roadsides or near industrial emitters, due to limited horizontal resolution. It is expected that dilution and coagulation during atmospheric transport from these hotspots to the surroundings will quickly reduce the UFP concentrations to levels that are realistically represented by the urban background (e. g. Karner et al., 2010, finding a distance of 1000 m for roadways). This is in agreement with a study from Salma et al. (2014), showing measured UFP concentrations at different locations across Budapest, Hungary. They measured on average a factor of 3 higher UFP concentrations in the city centre compared to the urban background, and an additional factor of 2 higher concentrations in street canyons.

#### 4.1.2   North America

**Table 2.** Summary of observations of PNCs (annual averages, $cm^{-3}$) in urban regions across the United States for different years taken from a study by Saha et al. (2021). The geographical location is indicated by latitude (Lat) and longitude (Lon). The annual averages of observed ($PNC_O$) and modelled ($PNC_M$) PNCs (2015 for $PNC_M$, variable years for the observations) for the grid box encompassing the station are compared. The measurements applied a cut-off minimum diameter (Cut). The same cut-off was used to calculate simulated $PNC_M$. Simulated PNCs after the emission sector-based redistribution are displayed as well ($PNC_R$), discussed in Sect. 4.2.

| Location | Lat | Lon | Year | Cut [nm] | $PNC_O$ | $PNC_M$ | $PNC_R$ |
|---|---|---|---|---|---|---|---|
| Rochester (NY) | 43.14 | -77.54 | 2013 | 10 | 4450 | 3799 | 6294 |
| Boston (MA) | 42.33 | -71.10 | 2016 | 10 | 12200 | 4614 | 9169 |
| Somerville (MA) | 42.40 | -71.09 | 2009 | 6 | 10000 | 4687 | 8640 |
| Queens (NY) | 40.73 | -73.82 | 2009 | 20 | 8210 | 4787 | 12722 |
| Long Island (NY) | 40.74 | -73.58 | 2009 | 20 | 7600 | 4787 | 9975 |
| Livermore (CA) | 37.69 | -121.78 | 2012 | 7 | 8220 | 6090 | 6553 |
| Red Wood (CA) | 37.48 | -122.20 | 2015 | 7 | 11910 | 6090 | 9757 |
| San Pablo (CA) | 37.96 | -122.36 | 2012 | 7 | 10480 | 6090 | 12119 |
| Santa Rosa (CA) | 38.44 | -122.71 | 2012 | 7 | 8660 | 6090 | 11526 |
| Anaheim (CA) | 33.83 | -117.94 | 2016 | 7 | 13950 | 8019 | 12100 |
| Central LA (CA) | 34.07 | -118.23 | 2016 | 7 | 17780 | 8019 | 14929 |
| Compton (CA) | 33.90 | -118.21 | 2012 | 7 | 14000 | 8019 | 14916 |
| Rubidoux (CA) | 34.00 | -117.42 | 2016 | 10 | 12930 | 7975 | 11858 |

In North America only measurements of PNCs are available. Annual averages of observed PNCs ($PNC_O$) at urban sites in the United States are taken from a study by Saha et al. (2021). The measurements were performed in different years, from 2009 to 2016. The results are summarized in Table 2. We note that the model again underestimates $PNC_O$ at central urban stations, mostly by a factor ranging from 1.5 to 2.5.

Additional observations of $PNC_O$ in the United States and Canada were obtained from the EBAS database providing daily measurements. They are compared to annual averages of the simulation for days with valid measurements, shown in Table 3. $PNC_M$ in Bondville, Egbert and at the Appalachian State University (ASU) agree reasonably well with the observations with respect to the aforementioned criteria (see Sect. 4.1.1), while at Southern Great Plains (SGP) and Trinidad Head the concentrations are overestimated by the model. The SGP site is located in the middle of wheat fields and pastures, and is thereby efficiently shielded from major UFP sources in the encompassing grid box. The Trinidad Head measurement was performed directly at the Californian Coast, and the corresponding grid box is influenced by a mixture of anthropogenic, rural and oceanic influences, leading to strong UFP gradients (comparable to Finokalia in Europe).

**Table 3.** Summary of observations of $PNC_O$ ($cm^{-3}$) across the United States and Canada in 2015, with data archived in the EBAS database. The geographical location is indicated by latitude (Lat) and longitude (Lon), the elevation of the site above the sea surface is indicated by the altitude asl in metres (Alt). The annual averages of $PNC_O$ and $PNC_M$ for the grid box encompassing the station are compared and different measures of agreement are listed. The same comparisons are performed after the redistribution of the model ("Redistributed model"), which will be discussed in Sect. 4.2. All included measurement stations contained at least 250 days of valid measurements in 2015. There is no particle size cutoff value given in the datasets, and thus none is applied on the simulation. Abbreviations: ASU – Appalachian State University, SGP – Southern Great Plains

| | Observations | | | | Model results | | | | Redistributed Model | | | |
|---|---|---|---|---|---|---|---|---|---|---|---|---|
| **Station** | Lat | Lon | Alt [m] | $PNC_O$ | $PNC_M$ | PF2 | NRMSE | $\overline{M/O}$ | $PNC_R$ | PF2 | NRMSE | $\overline{M/O}$ |
| ASU, Boone (NC, US) | 36.21 | -81.69 | 1076 | 2930 | 2471 | 58 | 0.26 | 0.81 | 3317 | 55 | 0.3 | 1.09 |
| Bondville (IL, US) | 40.05 | -88.37 | 213 | 4095 | 4130 | 55 | 0.26 | 0.96 | 4343 | 55 | 0.27 | 1.01 |
| Egbert (ON, CA) | 44.23 | -79.78 | 255 | 4962 | 3238 | 53 | 0.26 | 0.68 | 2775 | 49 | 0.27 | 0.58 |
| SGP E13 (OK, US) | 36.60 | -97.48 | 318 | 3600 | 6966 | 40 | 0.61 | 1.82 | 7457 | 38 | 0.67 | 1.95 |
| Trinidad Head (CA, US) | 41.05 | -124.15 | 107 | 1526 | 2882 | 48 | 0.59 | 1.8 | 2800 | 49 | 0.57 | 1.75 |

### 4.1.3 India

**Table 4.** Summary of $UFP_O$ concentrations ($cm^{-3}$) derived from PSDs in India for different years. The geographical location is indicated by latitude (Lat) and longitude (Lon). The annual averages of $UFP_O$ and $UFP_M$ concentrations (2015 for $UFP_M$, variable years for the observations) for the grid box encompassing the station are compared. The measurements applied a cut-off minimum diameter (Cut). The same cut-off was used to calculate simulated $UFP_M$. Simulated UFP concentrations after the emission sector-based redistribution are displayed as well ($UFP_R$), which will be discussed in Sect. 4.2. Abbreviations: IITM – Institute of Information Technology & Management, IITD – Indian Institute of Technology Delhi, MRIU – Manav Rachna International University

| Station | Lat | Lon | Year | Cut [nm] | $UFP_O$ | $UFP_M$ | $UFP_R$ | Reference |
|---|---|---|---|---|---|---|---|---|
| Delhi IITM | 28.61 | 77.1 | 18/19 | 14.1 | 19750 | 15339 | 28789 | Thamban et al. (2021) |
| Delhi IITD | 28.55 | 77.19 | 17/18/19 | 14.1 | 42972 | 15339 | 24917 | Thamban et al. (2021) & Gani et al. (2020) |
| Delhi MRIU | 28.45 | 77.28 | 18/19 | 14.1 | 17350 | 15339 | 15959 | Thamban et al. (2021) |
| Mahabaleshwar | 17.92 | 73.66 | 15 | 5.5 | 2441 | 2248 | 2316 | Sebastian et al. (2022) |
| Hyderabad | 17.46 | 78.32 | 19/20/21 | 10 | 4680 | 6350 | 11131 | Sebastian et al. (2022) |

Annual average $UFP_O$ concentrations in Delhi (Thamban et al., 2021), Mahabaleshwar and Hyderabad (both from Sebastian et al., 2022) were obtained from a collaboration with groups performing field measurements. Additionally, we adopted daily measurements of PSDs from Gani et al. (2020) for the Indian Institute of Technology Delhi (IITD), which we converted to

UFP concentrations and combined with the measurements of Thamban et al. (2021). The evaluation for India is summarized in Table 4.

The simulation grid cell covering Delhi underestimates $UFP_O$ concentrations at all three urban measurement stations in Delhi. The higher $UFP_O$ concentrations over the Indian Institute of Technology Delhi (IITD) compared to the two other measurements may be due to its proximity to a major highway. In contrast, our simulations are high-biased over Hyderabad. The measurement station in Hyderabad is a suburban university campus, approximately 15 km from the city centre, where UFP concentrations are expected to be significantly reduced compared to the downtown regions with strong traffic emissions.

The consequent UFP gradients may not be adequately resolved by our model calculations (spatial representation error after Schutgens et al. (2016a)).

### 4.1.4 China

**Table 5.** Summary of $UFP_O$ concentrations ($cm^{-3}$) derived from PSDs in China for different years. The geographical location is indicated by latitude (Lat) and longitude (Lon). The annual averages of $UFP_O$ and $UFP_M$ concentrations (2015 for the simulation, variable years for the observations) for the grid box encompassing the station are compared. The measurements applied a cut-off minimum diameter (Cut). The same cut-off was used to calculate simulated $UFP_M$. Simulated UFP concentrations after the emission sector-based redistribution are displayed as well ($UFP_R$), which will be discussed in Sect. 4.2.

| Station | Lat | Lon | Year | Cut [nm] | **$UFP_O$** | **$UFP_M$** | **$UFP_R$** | Reference |
|---|---|---|---|---|---|---|---|---|
| Shanghai | 31.17 | 121.43 | 14 | 13.4 | 12800 | 10850 | 23623 | Unpublished |
| Beijing | 39.94 | 116.29 | 18/19 | 5.6 | 14812 | 11489 | 20123 | Liu et al. (2020) |
| Beijing | 39.9 | 116.38 | 04–06 | 3.0 | 24900 | 11408 | 21041 | Wu et al. (2008) |
| Lin'an | 30.28 | 119.75 | 15 | 3.8 | 4928 | 11815 | 12447 | Shen et al. (2022) |
| Gucheng | 39.15 | 115.73 | 18/19 | 12.9 | 8284 | 10678 | 9680 | Li et al. (2021) |

    Observations of PSDs in China were obtained through collaboration with relevant groups for Shanghai (unpublished), Beijing in 2018/19 (Liu et al., 2020), Lin'an (Shen et al., 2022) and Gucheng (Li et al., 2021). Additionally, we included observa-

tions from Beijing from 2004–2006 (Wu et al., 2008). We compare the UFP concentrations derived from our simulation to the PSD measurements from these sites. The results are summarized in Table 5.

    $UFP_M$ concentrations in Shanghai and Beijing are slightly lower (approximately $20\%$) than $UFP_O$ concentrations for the years closest to 2015 (2014 for Shanghai, 2018/19 for Beijing), unlike other urban locations discussed above. However, $UFP_O$ concentrations in Beijing from 2004–2006 are underestimated by more than a factor of 2 by $UFP_M$. A probable reason for

that is the various air pollution reduction measures taken in China, especially in Beijing. In Beijing, $PM_{2.5}$ concentrations decreased from $89\,\mu g/cm^3$ in 2013 to $58\,\mu g/cm^3$ in 2017 and $42\,\mu g/cm^3$ in 2019 (Zeng et al., 2019; Lu et al., 2020), implying a simultaneous reduction of UFP sources. With the air pollution reduction measures industries also moved further away from city centres, potentially decreasing the association with UFP concentrations (see also Sect. 4.2).

UFP$_M$ overestimates UFP$_O$ strongly in Lin'an and slightly in the Beijing suburban Gucheng. Lin'an is influenced by the neighboring city of Hangzhou with approximately 10 million inhabitants, Gucheng by sharing the grid box with central Beijing.

### 4.1.5 Remote regions

**Table 6.** Summary of UFP$_O$ concentrations $(\mathrm{cm}^{-3})$, respectively PNC$_O$, in remote regions. Four different settings are distinguished, namely Ocean, Forest, Mountain and Polar. The geographical location is indicated by latitude (Lat) and longitude (Lon), the elevation of the site above the sea surface is indicated by the altitude asl in metres (Alt). The annual averages of the observed (Obs) and simulated (Mod) concentrations for the grid box encompassing the station are compared and different measures of agreement are listed (see Table 6). We applied the measurement detection limit on the simulation for all UFP concentrations, while information on the detection limit was not available for the PNC measurements.

| Station | **Setting** | Lat | Lon | Alt | Meas | **Obs** | **Mod** | PF2 | NRMSE | $\overline{\mathrm{M/O}}$ | Reference |
|---|---|---|---|---|---|---|---|---|---|---|---|
| ATom campaign | Ocean | - | - | < 200 | UFP | 403 | 800 | - | - | - | Brock et al. (2019) |
| ATTO Tower (Brasil) | Forest | -2.14 | -59.0 | 120 | UFP | 376 | 199 | - | - | - | Franco et al. (2022) |
| BEO Moussala (BG) | Mountain | 42.18 | 23.59 | 2925 | UFP | 542 | 940 | 45 | 0.51 | 1.58 | EBAS |
| Hyytiälä (FI) | Forest | 61.85 | 24.3 | 179 | UFP | 1430 | 1140 | 59 | 0.21 | 0.77 | EBAS |
| Sammaltunturi Pallas (FI) | Polar | 67.97 | 24.12 | 565 | UFP | 556 | 489 | 43 | 0.24 | 0.71 | EBAS |
| Trollhaugen (NO) | Polar | -72.02 | 2.53 | 1309 | UFP | 154 | 62 | 31 | 0.23 | 0.36 | EBAS |
| Värriö (FI) | Polar | 67.75 | 29.61 | 390 | UFP | 697 | 800 | 44 | 0.34 | 0.93 | EBAS |
| Zeppelin Mountain (NO) | Polar | 79.9 | 11.86 | 473 | UFP | 161 | 185 | 40 | 0.42 | 0.55 | EBAS |
| Zugspitze (DE) | Mountain | 47.41 | 10.98 | 2650 | UFP | 944 | 655 | 57 | 0.22 | 0.65 | EBAS |
| Alert (NU, CA) | Polar | 82.50 | -62.34 | 2182 | PNC | 205 | 279 | 47 | 0.26 | 1.48 | EBAS |
| Barrow (AK, US) | Polar | 71.32 | -156.61 | 11 | PNC | 277 | 417 | 33 | 0.44 | 0.98 | EBAS |
| South Pole | Polar | -90.00 | -24.80 | 2841 | PNC | 192 | 238 | 56 | 0.21 | 1.5 | EBAS |

We considered remote measurements over the open ocean, in forests, on mountains and polar sites, both of PNCs and UFP concentrations. The simulation and measurement results are summarized in Table 6. Simulated average concentrations are all within a factor of 2 of the observations. Especially the Northern Hemispheric polar regions show good agreement, while there is a stronger over-, respectively underestimation of UFP$_O$ at the two measurement stations in Antarctica, both at elevated altitude.

The annual average forest UFP$_O$ concentrations at the ATTO Tower in Brasil (Franco et al., 2022) and in Hyytiälä (EBAS database) are both underestimated by the simulation. The timeline analysis of the measurements in the Amazon forest (ATTO) shows this pronounced underestimation only in winter, i. e. UFP$_M$ shows a stronger seasonality than UFP$_O$. The potential causes will be subject of future studies, also considering data from an upcoming measurement campaign.

Measurements over the open ocean were taken from the ATom aircraft campaign (Brock et al., 2019), conducted in different seasons from 2016 to 2018. We collected all measurements that were performed below 200 m over the ocean and compared

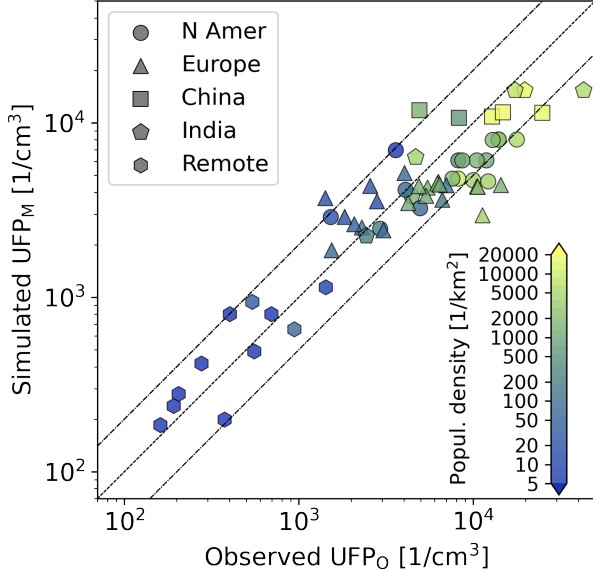

**Figure 7.** Summary of all measurement stations in a scatter plot showing observed UFP$_O$ concentrations on the x axis and simulated UFP$_M$ concentrations on the y axis. Different symbols indicate the different regions, respectively settings. The color represents the local population density, sampled at the measurement stations with a resolution of $0.1° \times 0.1°$.

them to daily averaged UFP$_M$ concentrations in the lowest vertical model level of the corresponding horizontal grid cell at the respective day of the year. UFP$_O$ concentrations over the Southern Ocean are mostly underestimated by the simulation, while concentrations over the Pacific are overestimated. Considering generally higher UFP$_M$ concentrations over the Pacific Ocean and lower UFP$_M$ concentrations over the Southern Ocean (compare Fig. 3), the open ocean UFP variance seems to be overestimated by the model.

### 4.1.6 Global

Figure 7 summarizes the evaluation results of all measurement stations presented and compares the annual averages of the observed to the simulated UFP concentrations in a scatter plot. The shapes of the symbols represent the five studied regions. Population density data (represented by the color scheme in Fig. 7) is taken from CIESIN, 2018, sampled at the measurement stations with a resolution of $0.1° \times 0.1°$.

The logarithms of UFP$_O$ and UFP$_M$ concentrations are correlated by $r = 0.93$ ($r = 0.76$ excluding remote regions), while the RMSLE (root mean square logarithmic error) is $0.55$ [1]. The geometric mean of the ratio between the modelled and observed mean values is $\overline{M/O} = 0.82$.

---

[1]We use the correlation and RMSE of the logarithmic values here as several orders of magnitudes are covered and the influence of the lower UFP concentrations would be negligible otherwise.

In remote areas all simulated concentrations are within a factor of 2 of the observed ones. For these areas the population density (and respective anthropogenic emissions) within the encompassing grid box is mostly uniform at low numbers of inhabitants, i.e. the grid cells only cover remote areas. Thus, the grid cell average of UFP concentrations is not influenced by densely populated and typically much more polluted regions.

All other measurement stations considered have in common that the encompassing grid boxes include an urban city centre with its surroundings, and thus have high variance in anthropogenic emissions within the grid box. Figure 7 indicates a link between local population density and UFP concentrations. In fact, the logarithm of the population density is positively correlated to the logarithm of the $UFP_O$ concentrations in non-remote regions with $r = 0.80$. Thus, it can be expected that the actual UFP concentrations in grid boxes encompassing urban regions and its surroundings is non-uniform, with higher UFP concentrations at higher population density and lower UFP concentrations at lower population density. This inability of in-situ observations to represent the grid box environment is defined as the spatial representation error and studied by Schutgens et al. (2016a), and was found to be the strongest close to sources in agreement with our evaluation results.

The spatial representation error is illustrated in Fig. 7. At low $UFP_O$ concentrations in non-remote regions the local population density is lower (background, suburban and rural stations in grid boxes including urban centres), while $UFP_M$ are higher than $UFP_O$ concentrations ($\overline{M/O} = 1.32$ for a population density smaller than $100\ \mathrm{individuals/km^2}$), as the model grid boxes are influenced by the urban regions. At higher $UFP_O$ concentrations population density increases as well and the simulation underestimates $UFP_O$ concentrations on average by a factor of almost 2 ($\overline{M/O} = 0.62$ for a population density in excess of $1000\ \mathrm{individuals/km^2}$), as these urban stations are surrounded by suburban and rural regions that lower the simulation grid cell average of $UFP_M$ concentrations.

The main cause of the spatial representation error is the limited model resolution of approx. $180 \times 180$ km (at the equator). Consequently, the correlation between the logarithm of the local population density and $UFP_M$ concentrations in non-remote regions is only $r = 0.57$, thus much lower than the correlation to $UFP_O$ concentrations. To overcome this, an increase in model resolution of at least a factor of 3 (in one dimension) would be necessary, which would lead to an unreasonably high demand of computing time. Thus, an alternative approach was developed to mimic a resolution increase, by retroactively (after the simulation) redistributing $UFP_M$ concentrations per grid box based on local high resolution primary anthropogenic emissions relative to the grid box averaged primary anthropogenic emissions, guided by observations. This will be discussed in the following section.

## 4.2 Downscaling based on primary anthropogenic emissions

Our results, in line with the observations, corroborate that UFP concentrations at the Earth's surface are strongly influenced by anthropogenic activity. Although our access to long-term measurements is limited, the data available from several urban stations show a high variance of $UFP_O$ concentrations among different sites in a city (e. g. for Leipzig, Germany and Los Angeles, USA), being enhanced when model grid boxes contain urban and rural environments (e. g. Salma et al., 2014). Population density and $UFP_O$ concentrations are highly correlated, while the evaluation of $UFP_M$ concentrations indicates discrepancies due to sharp gradients of anthropogenic emissions within areas covered by the model grids.

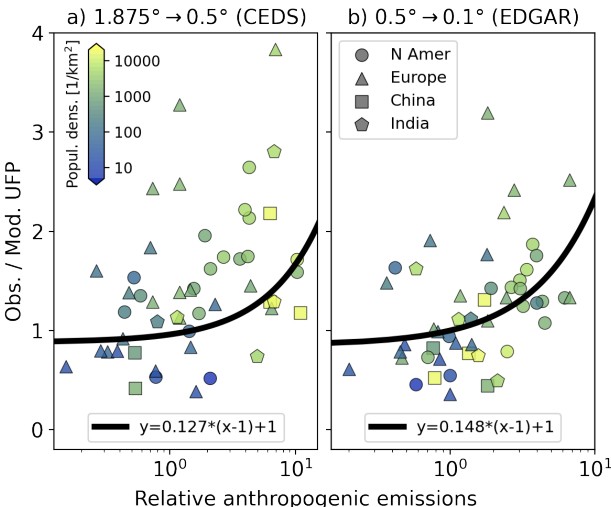

**Figure 8.** Scatter plot of the *relative anthropogenic emissions* (RAE) against the model-measurement discrepancy (Observed / Simulated UFP concentrations). RAE of the CEDS database relative to the model grid box are directly compared to the model-measurement discrepancy (a), while the RAE of CEDS relative to the CEDS grid box are compared to the remaining discrepancy after the first downscaling (b). We use the same color code for the population density and shapes for the regions as in Fig. 7. The black lines show a linear fit crossing the (1,1) point. Higher RAE correspond to more pronounced underestimation of the observations.

Primary anthropogenic emissions (PAE) from the CEDS database are available at a higher than model resolution of $0.5° \times 0.5°$ (hereafter referred to as $PAE_{CEDS;0.5}$) and were regridded to the simulation mesh (resulting in grid box averaged emissions $PAE_{CEDS;GB}$). Primary anthropogenic emissions from the EDGAR database v6.1 (Crippa et al., 2022) are available at even higher resolution of $0.1° \times 0.1°$ (hereafter referred to as $PAE_{EDGAR;0.1}$). Studies showed that locally enhanced UFP concentrations usually reach (urban) background levels within $1000\,\mathrm{m}$ from sources (Karner et al., 2010, e. g.), and the curb-

side UFP concentrations are highly localized. Hence, this section aims to use local $PAE_{CEDS;0.5}$ and $PAE_{EDGAR;0.1}$ and the relation to the respective grid box average at coarser resolution to fine-tune the simulation results guided by the observations, gaining improved resolution (downscaling) and closer agreement with observations (reducing the spatial representation error), especially in urban centres and their surroundings. It is important to mention that we do not modify the total number of $UFP_M$ per grid box in the following, but that $UFP_M$ concentrations are merely redistributed within each grid box.

Figure 9 illustrates the two-step downscaling procedure by the example of New Delhi and surroundings, using $PAE_{CEDS;0.5}$ to downscale to $0.5° \times 0.5°$ and $PAE_{EDGAR;0.1}$ for further downscaling obtaining a UFP dataset with $0.1° \times 0.1°$ horizontal resolution. As a basis we calculate *relative anthropogenic emissions* (RAE), i.e. the local anthropogenic particle number emissions relative to the average emissions of the grid box at coarser resolution:

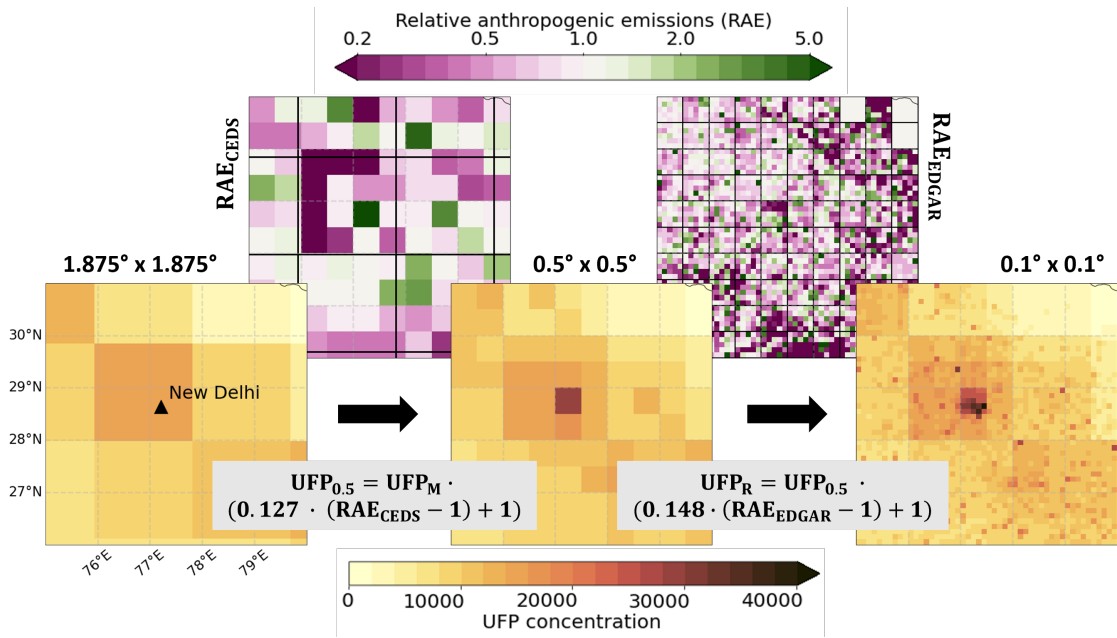

**Figure 9.** Illustration of the downscaling procedure using anthropogenic particle number emissions from the CEDS (Hoesly et al., 2018) and EDGAR (Crippa et al., 2022) emission databases by the example of New Delhi and surroundings. Relative anthropogenic emissions (RAE; local anthropogenic particle number emissions relative to the average over the initial grid box; displayed on the top) are set in relation to the model-observation discrepancy (Observed / Modelled UFP concentrations), displayed in Fig. 8. The resulting relation is used for downscaling UFP concentrations from model ($1.875° \times 1.875°$) to $0.5° \times 0.5°$ (CEDS) respectively $0.1° \times 0.1°$ (EDGAR) horizontal resolution in two steps. The average of the RAE over each grid box of the coarser resolution (black boxes) equals to 1, and thus the respective average UFP concentrations per grid box remain unchanged. Note that the color scale for UFP concentrations is differing from Fig. 3.

$$\mathrm{RAE_{CEDS} = PAE_{CEDS;0.5}/PAE_{CEDS;GB}}$$

$$\mathrm{RAE_{EDGAR} = PAE_{EDGAR;0.1}/PAE_{EDGAR;0.5}}$$

$\mathrm{RAE_{CEDS}}$ (top left in Fig. 9), respectively $\mathrm{RAE_{EDGAR}}$ (top right in Fig. 9), can be interpreted as the local excess or deficit of $\mathrm{PAE_{CEDS;0.5}}$ ($\mathrm{PAE_{EDGAR;0.1}}$) over $\mathrm{PAE_{CEDS;GB}}$ ($\mathrm{PAE_{EDGAR;0.5}}$). We limit $\mathrm{RAE_{CEDS}}$ to a maximum value of 10 and $\mathrm{RAE_{EDGAR}}$ to 7 due to missing observational datasets at locations with higher RAE.

In the first step we investigated the relationship between $\mathrm{RAE_{CEDS}}$ and the underestimation of $\mathrm{UFP_O}$ concentrations, re-
spectively $\mathrm{PNC_O}$, at each station ($\mathrm{UFP_O/UFP_M}$ or $\mathrm{PNC_O/PNC_M}$) for all evaluation results in grid boxes with anthropogenic emissions exceeding two million particles per square meter and second[2]. The relationship is displayed in Fig. 8(a). As expected there is a logarithmic correlation ($r = 0.42$) between the two quantities. We perform a linear fit that crosses the point (1,1),

---

[2]All high resolution pixels in the remaining (less anthropogenically influenced) grid boxes are bilinearly interpolated with respect to the coarser resolution.

i.e. $y = c_{\mathrm{CEDS}} \cdot (x - 1) + 1$, where y is the ratio $N_{obs}/N_{mod}$ and $x$ are the $\mathrm{RAE}_{\mathrm{CEDS}}$. This function was chosen as it is the only global function that conserves the average grid box $\mathrm{UFP}_{\mathrm{M}}$ concentrations after applying it on the model results. The fit
parameter $c$ was determined to be $c_{\mathrm{CEDS}} = 0.127$ using a logarithmic least squares fit. Thus, if $\mathrm{RAE}_{\mathrm{CEDS}}$ increases by 1, the model underestimation increases by 0.127. The fit function is shown as a black line in Fig. 8(a). This relationship is used to downscale simulated $\mathrm{UFP}_{\mathrm{M}}$ concentrations, yielding redistributed UFP ($\mathrm{UFP}_{0.5}$) concentrations.

This procedure is repeated for the relationship between $\mathrm{RAE}_{\mathrm{EDGAR}}$ and the remaining model-observation discrepancies after the first downscaling (Fig. 8(b)). We determine $c_{\mathrm{EDGAR}}$ to be $c = 0.148$, and thus $\mathrm{UFP}_{0.5}$ can be further downscaled to the
urban and industrial environments, yielding redistributed $\mathrm{UFP}_{0.1}$ (hereafter referred to as $\mathrm{UFP}_{\mathrm{R}}$) concentrations (rightmost plot in Fig. 9). Summarized, we obtain two downscaling functions that are applied sequentially to simulated $\mathrm{UFP}_{\mathrm{M}}$ concentrations:

$$\mathrm{UFP}_{0.5} = \mathrm{UFP}_{\mathrm{M}} \cdot (0.127 \cdot (\mathrm{RAE}_{\mathrm{CEDS}} - 1) + 1)$$
$$\mathrm{UFP}_{\mathrm{R}} = \mathrm{UFP}_{0.5} \cdot (0.148 \cdot (\mathrm{RAE}_{\mathrm{EDGAR}} - 1) + 1)$$

To demonstrate the independence of the downscaling procedure from the data used for the fit, we randomly subdivided the
non-remote measurement stations into a training and a test dataset (25 for training, 24 for testing) in 5000 different random ways, derived the fit parameters from the training dataset only, and applied them on the test dataset subsequently. The results are displayed in Fig. 10. The fit parameters $c_{\mathrm{CEDS}}$ and $c_{\mathrm{EDGAR}}$ range between 0.07 and 0.23 in $90\%$ of the runs, while the average values are very close to the values derived from the complete dataset, and all derived fit parameters are greater than 0. The RMSLE consistently decreases with rising order of downscaling, with an average improvement of 0.11 in total ($95.6\%$ of
the time RMSLE decreases), while the logarithmic correlation increases on average by 0.07 (improves $95.1\%$ of the time). The bias $\overline{\mathrm{M/O}}$ evolves from 0.78 on average in the model output resolution ($2.4\%$ of the time between 0.9 and 1.1) to 0.92 after the CEDS downscaling ($55.3\%$ between 0.9 and 1.1) to 1.04 ($66.2\%$ between 0.9 and 1.1) after the final downscaling, slightly overestimating the observed values. We conclude that the fit parameters can be applied to data points outside the training dataset. However, as we want to maximise the number of measurement stations to use for the global downscaling, we use the
complete evaluation results for the downscaling as described before.

The resulting $\mathrm{UFP}_{\mathrm{R}}$ concentrations are included in Tables 1–5 in the last columns. At measurement sites in Europe, urban regions in the United States, India and China the agreement with measurements generally improves. For instance, $\mathrm{UFP}_{\mathrm{R}}$ concentrations in Madrid, the measurement station with the strongest underestimation in this analysis (triangle in the top right corner in Fig. 8(a)), are increased by a factor of 2.1, strongly improving the agreement (see Table 1: PF2 from 22 to 59 %,
$\overline{\mathrm{M/O}}$ from 0.28 to 0.59). The remaining underestimation is likely caused by the influence of nearby roads with heavy traffic, which are of localized relevance only.

$\mathrm{UFP}_{\mathrm{R}}$ in Beijing and Shanghai are strongly increased by the redistribution as well, which can even lead to an overestimation of the observed $\mathrm{UFP}_{\mathrm{O}}$ concentrations (see Table 5). However, for Beijing the redistributed $\mathrm{UFP}_{\mathrm{R}}$ fall between the high $\mathrm{UFP}_{\mathrm{O}}$ from 2004–2006 and the lower $\mathrm{UFP}_{\mathrm{O}}$ from 2018/19. This is in line with the emission reduction in China (see Sect. 4.1.4).

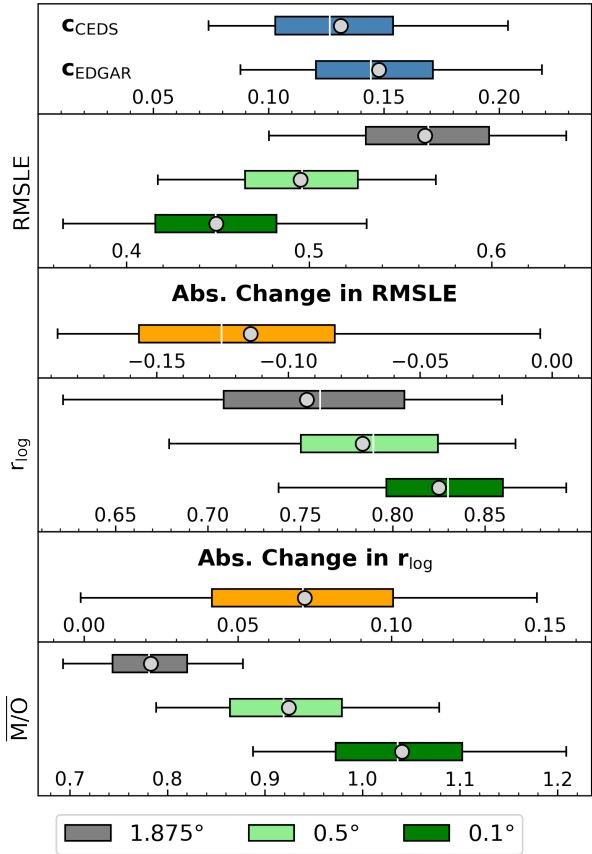

**Figure 10.** Results of 5000 independent training and test runs for the fit parameter estimation (25 stations in training dataset, 24 in test dataset). The fit parameters are displayed in the top panel, while the statistical measures are displayed below at different orders of downscaling, along with the absolute change of RMSLE and the logarithmic correlation. We show box-whisker plots, where the boxes illustrate the 25-75% interval and the whiskers the 5-95% interval. Median values are marked by the vertical white line and mean values by the light-grey circles.

$UFP_R$ concentrations over the Beijing suburban Gucheng, sharing the model grid box with Beijing, are reduced, approaching $UFP_O$ concentrations.

Similarly, $UFP_R$ concentrations are strongly increased at the measurement stations Delhi IITM and Delhi IITD, resulting in $UFP_R$ concentrations in-between the two $UFP_O$ concentrations. Moreover, $UFP_R$ is only slightly increased at Delhi MRIU due to lower local $PAE_{HR}$, remaining similar to $UFP_O$.

The left panel of Fig. 11 shows the comparison between $UFP_R$ and $UFP_O$ concentrations (analogously to Fig. 7) after the applied redistribution. In spite of the redistribution of UFPs within grid cells, there is still an overestimation of $\overline{M/O} = 1.27$ (1.32 before) at low population density below $100 \, \mathrm{individuals/km^2}$. On the other hand, in densely populated regions (more

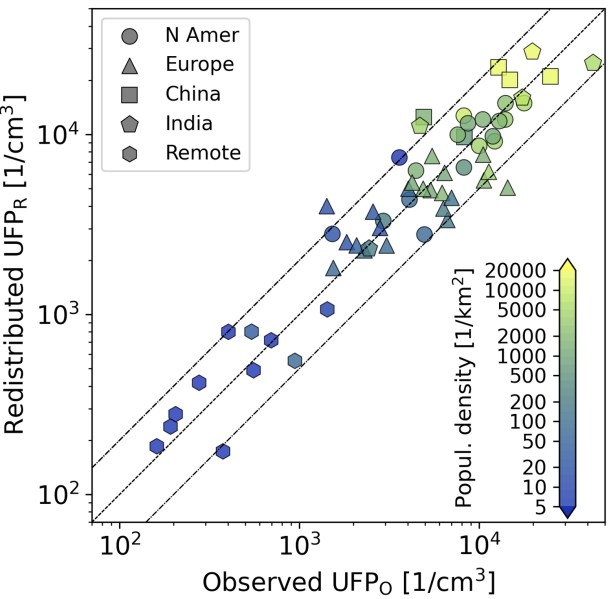
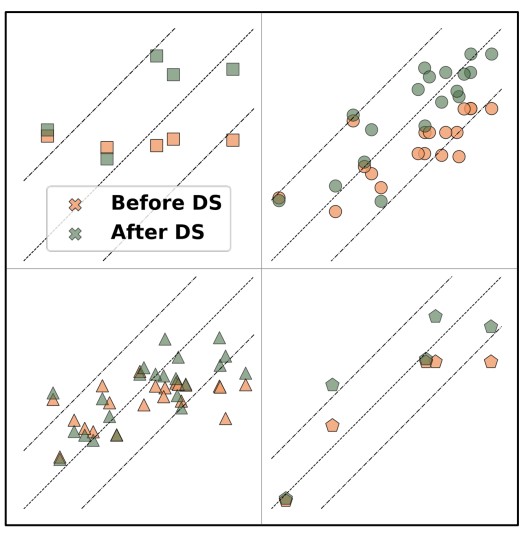

**Figure 11.** Left: Same as Fig. 7, but after the redistribution of $UFP_M$ concentrations (downscaling) based on the *relative anthropogenic emissions* per grid box (more details in the text). Right: Evaluation results before and after the downscaling (DS) for China, North America Europe and India (from top left to bottom right). Note that the value range differs in all subpanels.

than 1000 $\mathrm{individuals/km^2}$) simulated and observed UFP concentrations are of the same magnitude ($\overline{M/O} = 1.01$). Thus, the biases described in Sect. 4.1.6 are strongly reduced though not fully eliminated, especially for scarcely populated regions.

Evaluation results before and after the downscaling are displayed in the right panel of Fig. 11. The strong improvement in municipal regions is apparent when considering the urban measurement stations in the United States (top right), aligning closely around the center line. An increase in correlation is also clearly visible in China, Europe and India.

  All $UFP_R$ concentrations differing by more than a factor of 2 from $UFP_O$ can be attributed to nearby major traffic nodes and motorways (Leipzig-Eisenbahnstrasse), or alternatively coastal sites (Finokalia), a shielded location in a wheat field (Southern

Great Plains E13) and influence of a neighbouring megacity (Lin'an and Hyderabad). These discrepancies between $UFP_R$ and $UFP_O$ may perhaps be resolved with very high ($< 1\,\mathrm{km}$) horizontal resolution, which is computationally impracticable with a global model and cannot be achieved with downscaling due to missing emission datasets with respective horizontal resolution.

  Finally, the logarithmic correlation between population density and $UFP_R$ concentrations increases to $r = 0.77$ ($r = 0.57$ for $UFP_M$) after redistribution, which is similar to the logarithmic correlation between population density and $UFP_O$ concentrations

($r = 0.80$). This indicates an improved representation of the anthropogenic influence in the downscaled dataset.

  Figure 12 shows $UFP_R$ concentrations in Asia after the applied downscaling, revealing more detailed features in anthropogenically influenced regions, especially in Eastern China and Northern India. Note that the color scale is extended as $UFP_R$

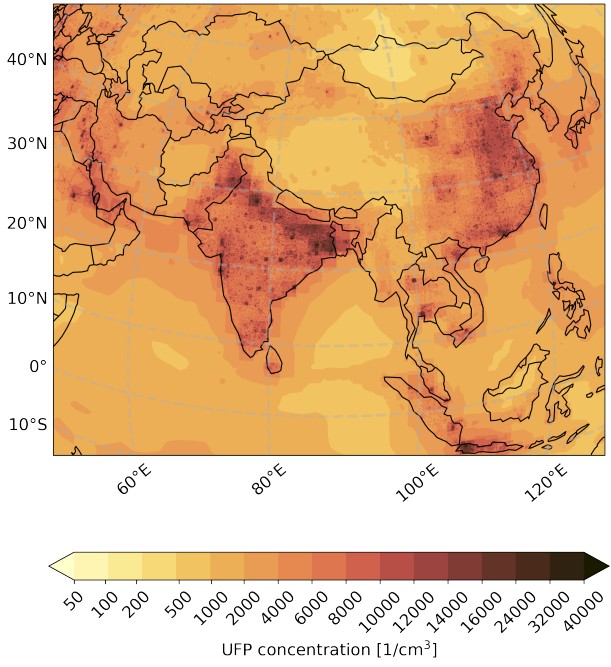

**Figure 12.** Annual average of UFP$_R$ concentrations in Asia simulated with EMAC for the year 2015 after the downscaling based on primary anthropogenic emissions at a resolution of $0.1° \times 0.1°$. Note that the color scale from Fig. 7 is extended as UFP$_R$ concentrations can exceed 40,000 per $\text{cm}^3$.

concentrations can exceed 40,000 per $\text{cm}^3$ in Indian industrial regions, as well as parts of urban environments in Mumbai, New Delhi, Shanghai, Riyadh, Kuwait and Kairo.

## 5   Limitations and Uncertainties

As the resulting UFP$_M$ and UFP$_R$ concentrations are an intricate interplay of emission parameters, numerical simulation, evaluation and observation-guided redistribution, it is not possible to directly infer quantitative uncertainties for the provided datasets. However, the final results in Fig. 11 show that $90\%$ of the annual averaged UFP$_R$ concentrations are within a factor of 2 of UFP$_O$ concentrations ($84\%$ for UFP$_M$) and all simulated UFP$_R$ concentrations are within a factor of 3 of the observations after the emission-grid based redistribution. Additional limitations and uncertainties will be discussed qualitatively next.

Median diameters of directly emitted particles per sector were estimated based on the emission size distributions from Paasonen et al. (2016) and measurement reports in the literature with associated uncertainties. Additionally, we assumed the diameters to be globally identical for each sector, and used the same diameters for all species. These simplifications lead to limitations in the precision by which the different sectors and species contribute to the total UFP$_M$, respectively UFP$_R$

number. Anthropogenic emissions are distributed in each time step based on monthly averages from the emission datasets. This potentially introduces bias, due to the non-linearity of chemical and microphysical processes. Future studies will include emission time factors, to improve the time resolution of emissions.

Apart from UFP concentrations, we also used PNCs for the evaluation. PNCs are a proxy for UFP concentrations, as UFPs tend to dominate the total particle number (e. g. Baldauf et al., 2016; Kumar et al., 2014). However, locally there can be deviations. To increase the number of measurements for the evaluation we additionally used observations from years that differ from the simulation year (2015). Different meteorological conditions and emissions potentially lead to biases in the evaluation. The downscaling was performed using evaluation results from differing years as well, while the simulation results were only taken from 2015. Thus, potential interfering effects may arise from the downscaling and general temporal trends as seen from the analysis of the interannual variability in Sect. 4. The interannual variability (up to 50 % at low absolute UFP concentrations and below 20 % over polluted regions over a timespan of 3 years) has been attributed to meteorological variations (along the ITCZ and mostly over the oceans), and decreasing (South and East Asia, Europe and the United States) or increasing (Nigeria and South Africa) long-term trends.

We define UFPs as all particles with a diameter smaller than 100 nm in the model without a lower cut on the particle diameter. Particles with a diameter below 1.7 nm are typically regarded as molecular clusters, and nucleation rates are reported as the frequency that clusters reach this threshold (e. g. by the CLOUD experiment; Kirkby et al., 2011). As a consequence, freshly formed particles are introduced to the model with a diameter of 1.7 nm. However, due to the log-normal modal aerosol distribution (Pringle et al., 2010), smaller particles down to infinitesimal small diameters are automatically created by definition. As a result, small fractions (up to around 2 % in polluted regions, with maximum values of up to 20 % locally over the oceans) of the presented UFPs have a diameter smaller than 1.7 nm. However, these particles are not considered as molecular clusters as the low diameter is a result of computational effects.

In addition to the uncertainties introduced by the horizontal resolution of the model, the vertical resolution of the model, ranging from $45 - 70$ m for the surface layer, adds additional uncertainties. While the surface layer is mostly entirely located in the boundary layer, decreases in UFP concentrations are still observed in general, as reported by Du et al. (2017); Harrison et al. (2019); Heintzenberg et al. (2011); Villa et al. (2017), with the strongest decrease with altitude being observed close to UFP sources. We analysed the relative change in UFP concentrations from the surface layer to the next-highest model level, exhibiting a decrease of 20-30% in UFP concentrations on average over polluted regions, with the reduction slightly exceeding 50% over Central Africa. At higher latitudes, we even see increasing UFP concentrations at the second lowest model level. The decrease over polluted regions is in line with the observations from Du et al. (2017); Harrison et al. (2019); Heintzenberg et al. (2011), who observe a decrease of up to 30 % at comparable altitudes[3]. Considering a linear decrease respectively increase within the range of 30% within the model surface layer, this adds an additional uncertainty of $\pm 15\%$ to the simulated UFP concentrations.

---

[3]Villa et al. (2017) even observe a decrease of up to 80%, however these measurements were performed directly at a highway and at some distance to other sources, which is not representative for a grid box covering approx. 180 x 180 km and overestimates the effect

Finally, an important uncertainty is the spatio-temporal representativeness of the observations with respect to the model grid box and time sampling. We report three different types of representation errors:

1. Purely spatial representation errors only due to the extent of the grid box at stations, for which timeline measurements of 2015 (mostly hourly) were available and we collocated our simulation with the observations according to Schutgens et al. (2016b), i. e. all observations from the EBAS database.

2. Spatio-temporal representation errors for measurement stations, for which only annual averages were available or the years were differing from the evaluation year (2015).

3. (Spatio-)temporal representation errors for the aircraft measurements from ATom, for which we used daily averaged model output and the measurement years are differing from the simuation year. The spatial representation error is reduced by the fact, that we have several measurements for each grid box.

The horizontal downscaling of UFP concentrations addresses and reduces the spatial representation error for types 1 and 2, while the temporal representation error of types 2 and 3 cannot be addressed.

The analysis in Sect. 4.2 showed that even after downscaling discrepancies in the evaluation remain, which are at least partly related to the representativeness of the measurement locations for the modelled grid areas (Schutgens et al., 2017, 2016a), also for $UFP_R$. Model concentrations of UFPs in rural regions in the vicinity of urban centres tend to be overestimated by the model (i.e. $UFP_R$). Hence, even for $UFP_R$ the horizontal resolution is still a limiting factor. To further reduce the spatial representation error, the simulations would need to be performed at very high resolution requiring currently impracticable computing resources, at least for the global scale, as well as additional measurement data for UFPs and meteorological parameters in urban and industrialized regions including roadsides and background urban environments. Salma et al. (2014) showed that UFP concentrations can typically vary by a factor of six from the urban background to street canopies within cities, and Karner et al. (2010) showed that UFP concentrations reach (urban) background concentrations within about a kilometre. A next step could be to apply high-resolution dynamical downscaling of concentrations guided by comprehensive measurements of which data may become available in the future, either using machine learning methods or a combination of operational street pollution models and human exposure modelling as done by Ketzel et al. (2021). Clearly, to make progress a much larger number of stations that continuously measure aerosol size distributions will need to be implemented.

## 6 Conclusions

We presented a first numerical simulation of ultrafine particles (UFPs) at the Earth's surface with the global EMAC model which includes a relatively detailed representation of aerosol formation and growth processes (i.e. the nucleation and Aitken aerosol size modes). Total concentrations were shown along with interannual variability in the years 2015-2017 and the dominating particle size segments around the globe. Emissions of gaseous and aerosol species were taken from the CEDS and CAMS databases, and the emission radii for aerosol species were taken or derived from the literature referring to the contributing source sectors.

Simulated UFP and particle number concentrations (PNCs) were evaluated using particle size distributions and PNCs from field measurements, the EBAS database, literature and published datasets. We generally achieve reasonable agreement between observed and simulated UFP concentrations, with good agreement for remote regions (forest, mountain, polar and ocean) that are not directly influenced by urban and other strong source regions (all simulated concentrations within a factor of 2 of the observations). In grid boxes with a high variance in population density we obtain larger deviations with observations related to the coarse model resolution (approximately 180 x 180 $km$ at the equator). UFP concentrations in urban regions with a high population density are underestimated by the simulation, while being overestimated in less densely populated regions. This representation error is associated with the high correlation between local population density and observed UFP concentrations (logarithmic correlation of $r = 0.80$), and is studied in depths by Schutgens et al. (2016a, b, 2017).

The relationship between the underestimation of the observed UFP concentrations and the local high resolution anthropogenic emissions relative to the grid box average was used to redistribute UFPs within each grid box, leaving the total number of particles unchanged. This yields a higher resolution, i. e. downscaled data for grid boxes with dominant anthropogenic influence, increases the agreement between observations and simulations (logarithmic correlation improves from 0.76 to 0.84 for non-remote regions, Root Mean Squared Logarithmic Error from 0.57 to 0.43), and decreases the spatial representation error by improving the representation of the anthropogenic impact on UFP concentrations.

We provide two global annual averaged datasets of UFP concentrations for the years 2015-2017 at different horizontal resolution that can be used for several purposes. The first dataset is given at a resolution of $1.875° \times 1.875°$ (roughly 180 x 180 km at the equator) and is directly derived from the simulation. This dataset can serve the purpose of global scale analyses of UFP concentrations, e. g. the comparison of different source regions, meteorology and atmospheric chemistry. The downscaled dataset has a much finer resolution of $0.1° \times 0.1°$ (roughly 10 x 10 km at the equator, and about 10 x 8 km at mid-latitudes) and includes the within-grid box UFP redistribution based on anthropogenic emission data. The latter is recommended to be used to characterize the exposure to UFPs in public health studies with a focus on densely populated regions, in particular the urban environment. Additionally, we make the 2015 UFP concentrations available in three size bins (1.7-20 nm, 20-50 nm, and 50-100 nm).

Future applications may also include studies on seasonality, anthropogenic source sectors and chemical composition of UFPs, and their contribution to health impacts of fine particulate matter.

*Code availability.* The Modular Earth Submodel System (MESSy) is continuously further developed and applied by a consortium of institutions. The usage of MESSy and access to the source code is licensed to all affiliates of institutions that are members of the MESSy Consortium. Institutions can become a member of the MESSy Consortium by signing the MESSy Memorandum of Understanding. More information can be found on the MESSy Consortium Website (http://www.messy-interface.org, last access: 17 February 2023; MESSy, 2023). The code presented here is available as git commit #49a7a544 in the MESSy repository, and all changes have been included in the main repository.

*Data availability.* We provide datasets with annual averages of UFP concentrations for the years 2015-2017, both in model resolution ($1.875° \times 1.875°$) directly derived from the model output, and at a resolution of $0.1° \times 0.1°$ with the observation-guided downscaling based on anthropogenic emissions. Additionally, we provide number concentrations in 3 size bins, i. e. 1.7-20 nm, 20-50 nm and 50-100 nm. The datasets are publicly available at [will be filled upon acceptance of the manuscript].

*Author contributions.* SC, AP and JL planned the research. AP prepared the model set-up and performed the simulation with the help of MK and SC. DS, YC, SNT, MS, GP and HW helped in collecting and/or directly provided the observational datasets. MK analysed the model results, evaluated the simulation, developed the downscaling procedure and wrote the manuscript with the help of AP. JL and AP supervised the project. All authors discussed the results and contributed to the review and editing of the manuscript.

*Competing interests.* At least one of the (co-)authors is a member of the editorial board of Atmospheric Chemistry and Physics.

*Acknowledgements.* MK acknowledges the financial support of the Max Planck Graduate Center with the Johannes Gutenberg University (Mainz). Hyderabad measurements were carried out with the financial support from Science Engineering Research Board, Government of India (ECR/2016/001333). MS acknowledges the Institute of Eminence, University of Hyderabad (sanction no. UoH/IoE/RC1/RC1-20-014). We acknowledge the work of Leslie Kremper, Marco Franco, Florian Ditas, Paulo Artaxo and Christopher Pöhlker, who collected and provided the observations at the ATTO tower. We additionally acknowledge the effort of Chao Yan and Xiaojing Shen for providing

the observational datasets from Beijing and Lin'an. The model simulations have been performed at the German Climate Computing Centre (DKRZ) through support from the Max Planck Society. Scientific colour maps (Crameri, 2021) are used in this study to prevent visual distortion of the data and exclusion of readers with colour vision deficiencies (Crameri et al., 2020).

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
