# Peer review of "Numerical simulation and evaluation of global ultrafine particle concentrations at the Earth's surface"

_EGUsphere, 2023_

## Author Comment (AC1)

**Referee comment #1:**

A global simulation of ultrafine particle concentrations for the year 2015 is presented and evaluated, with a technique to infer concentrations at high spatial grid resolution from a coarser resolution global model based on a high resolution emission inventory. The primary motivation is health effects.

The chemical transport model is state-of-the-art, as is the emissions data: EDGAR data for 2015 were only recently released at the high resolution used. However, the grid resolution of the model is surprisingly low (see comment below). An impressive number of evaluation datasets are used. The model performs well in the evaluation, though many discrepancies and details are surely lost in the annual averages presented. The paper is well written.

I recommend the paper for minor revisions, though the editor will want to consider that my suggestions are on the boundary between minor and major: calculations will take time and I suggest a new figure be added.

We thank the reviewer for reviewing our manuscript and for providing helpful comments. We report the comments (grey, bold) along with our replies (black).

**1) Introduction: a few more state-of-the-art papers on global aerosol number concentration, for example by Liu and Matsui, `https://doi.org/10.1029/2022GL100543`, and Chen et al, `https://acp.copernicus.org/articles/21/9343/2021/`, should be cited and discussed.**

We thank the reviewer for the additional literature. We added the references in the Introduction (along with supplemetary references, especially on vertical distributions of particle number concentrations and particle size distributions). We also discussed the results of Chen et al. (2021) in more detail in the results section, pointing out good agreement in most continental regions, while they simulated considerably lower PNCs in India compared to our simulated UFP concentrations (which are supported by observations). We did not focus on the work of Liu and Matsui (2022), as it mostly concentrates on (and evaluates the results against) vertical distributions, which is not the scope of our manuscript.

**2) I think the duration of the simulation or the time period for which the data are applicable should be mentioned in the abstract or early in the introduction, and maybe again when nudging is mentioned. It would also be useful to specify the time resolution of the model output that was used in the evaluation, and the time resolution of the dataset that will be published, in the abstract or introduction, as this will help potential users of the dataset (some might want diurnal or weekly cycles, for example).**

We added the duration of the simulation (which was extended to the years covering 2014-2017 with 2014 as a spin-up year) in the abstract, the introduction and also when introducing the nudging. We mentioned the time resolution of the output in the abstract ("annual averaged") and in the Data availability statement. We additionally added the information in the conclusions. The time resolution (daily averages) of the model output that was used for the evaluation was also added in the evaluation part. We do not plan to publish diurnal or weekly cycles, or

any other higher time-resolved data here, as this study focuses on long-term concentrations, and we use monthly averaged emissions (we are aware of the uncertainties introduced by that - see reply to comment 7). We plan to study seasonality and diurnal patterns in more detail in the future, by using emission time factors to distribute the emissions more realistically.

**3) The grid resolution is coarser, in both vertical and horizontal, even than that of a good number of CMIP6 climate models (that were run for hundreds of years tens of times). This seems odd since only two years were simulated. Surely, despite the high complexity of the chemistry and aerosol model, simulating two years at this resolution did not take more than a few days using a modest HPC resource at DKRZ? And only one simulation is shown – there are no sensitivity studies (unlike in the study of Gordon et al 2017 the authors cite, which also had low resolution). 1 degree resolution and 60 levels is typical of CMIP6 Earth System models, which surely are not more than a factor ten cheaper per simulated year unless EMAC is very inefficient or unnecessarily complex (e.g. > 10 volatility bins, or thousands of chemical species). Something to discuss? I should point out that the authors' work to infer higher resolution output using the emissions datasets is still clearly necessary and valuable despite this comment.**

Climate models used for CMIP6 are optimized to run on longer timespans, and do not include extensive and computationally demanding chemistry and aerosol schemes. Our chemical mechanism comprises more than 100 gas phase species and more than 250 reactions (Jöckel et al., 2006). Additionally, we include aerosol microphysics and explicit aerosol-gas equilibrium calculations (Pringle et al., 2010; Tsimpidi et al., 2014). A typical simulation of 2 years takes 4 days on 3 *AMD 7763 CPU* nodes with 128 cores each at the German Climate Computing Centre (DKRZ, 2021). An increase of horizontal and vertical resolution of a factor of 2 would additionally require an increase of the model timestep by roughly a factor of 2 due to the Courant-Friedrichs-Lewy condition (Courant et al., 1967). This would lead to a model runtime of approximately 2 months for the two simulated years. As we decided to increase the duration of the model run to 4 years (2014-2017) based on the comments of the referees, this would even increase to four months.

On the other hand, we do not expect a drastic increase in agreement with the model, as the resolution of $1.0° \times 1.0°$ is still too coarse to resolve urban to rural contrasts and would require further downscaling (as the reviewer also pointed out), and the vertical extent of the surface layer is identical for the higher resolution vertical resolution setups available for our model (with minor improvements in resolution for the following model levels). Sensitivity studies for different emission sectors were performed for a follow-up study on the health effects and are not the scope of this manuscript, but would additionally drastically increase the computational costs with an increased resolution.

We therefore decided not to increase the resolution at that point, but might do so in future studies. We also added this reasoning to the manuscript.

**4) Table 1 and 6: for mountain sites, did the authors compare the lowest model level with observations, or calculate the level that matches the altitude of the site relative to the average surface altitude in the 180x180km grid box?**

We did the latter, sampling the UFP concentrations at the level that matches the site altitude. We state this in line 234/235: "UFP$_M$ concentrations were sampled at the vertical grid box

covering the measurement site altitude."

5) Table 6: how was the comparison with aircraft measurements done? By interpolating, or matching grid cells, with daily or monthly averaged model output, to the four days (in the case of ATom) in which any given grid cell was sampled? A discussion of representativeness uncertainties would be appropriate here (see papers by Schutgens et al, e.g. https://acp.copernicus.org/articles/17/9761/2017/, which could also nicely set the scene for the downscaling work).

[Figure]

Figure 1: Summary of ATom measurements below 200m asl, compared to simulated values at the same day of the year in the relevant model grid box. Regions (Northwest pacific - NW Pac, Southwest Pacific - SW Pac, Southern Ocean - S Ocean, South Atlantic - S Atl, North Atlantic - N Atl) and seasons were distinguished by color scale and shape. Note that the timing of measurements and model calculations does not coincide (differing years), implying the meteorological variability in UFP concentrations is poorly captured.

We did the comparison by matching grid cells and the respective days when the data was sampled. We included a more detailed analysis in Fig. 1 of this document. We acknowledge two main uncertainties: Firstly, measurements and observations are from different years, and thus meteorological conditions do not coincide. Secondly, we compare daily averaged model output against the observations, which leads to representation errors according to Schutgens et al. (2017). A more extensive evaluation of our model against aircraft campaigns, including ATom, will be subject of future work, after finalising the model setup for the free and upper troposphere. We added a brief summary of the mentioned uncertainties to the "Limitations and Uncertainties" section.

6) Excluding free-troposphere stations from the evaluation on the grounds that the model wouldn't perform well there because it is optimized for the boundary layer seems odd. What optimization was done? Do the authors have reason to believe the model will perform badly in the FT, and does this also have implications for the boundary layer?

We apologise for formulating this in a misleading way. The simulation was not specifically "optimized" for the boundary layer, but rather the focus of the study is the boundary layer. We are currently working on extending the setup and also evaluating it against measurements in the free troposphere and the stratosphere. High altitude measurements at mountain peaks are strongly influenced by the steep orography, which our model cannot represent, and at the same time is not the scope of the manuscript. Potential issues in the free and upper troposphere can influence boundary layer processes as well. However, the evaluation focuses on the boundary layer, and we show good agreement there especially after the downscaling. We reformulated this description.

**7) What diurnal cycles for emissions were assumed? Weekly cycles? The nonlinearities in chemistry and microphysics probably make this quite important even if only monthly averages are considered.**

For this simulation we used monthly averages of emissions. We agree with the reviewer that this introduces additional uncertainties due to non-linearities, which we cannot reduce at the moment, and thank the reviewer for pointing that out. Thus, we added these uncertainties in the "Limitations and Uncertainties" section ("Anthropogenic emissions are distributed in each time step based on monthly averages from the emission datasets. This potentially introduces bias, due to the non-linearity of chemical and microphysical processes. Future studies will include emission time factors, to improve the time resolution of emissions."). We are working on an extension to the model setup to additionally include emission time factors (manuscript on that in preparation) for daily and diurnal variations, and plan to look into seasonal and diurnal cycles of UFP concentrations in more detail in the future.

**8) L105 may as well state the actual number of volatility bins.**

Added, alongside with the range of saturation concentration values: "ORACLE distinguishes between primary and secondary organic aerosols from different sources and volatilities (in up to 5 logarithmically spaced saturation concentration bins, ranging from $10^{-2}$ to $10^6$ ug/m$^3$, depending on the emission sector)."

**9) Figure 2: The nucleation mode geometric mean diameter is between 1 and 1.5nm in this case, so half the particles in the mode are smaller than 1.5nm, and a big fraction are smaller than 1nm. This is really small! The CLOUD NPF parameterizations that are used produce particles at 1.7nm diameter. So does it make sense to produce a dataset with nucleation-mode particles smaller than that? Since the authors use CLOUD NPF parameterizations, I would suggest the authors exclude anything smaller than the CLOUD collaboration's favourite cut-off diameter of 1.7nm from their model output and their evaluation tables and think about tweaking their model for future studies.**

We thank the reviewer for pointing out this issue. We agree that particles below 1.7 nm in diameter seem unreasonable to include, and being molecular clusters, as defined e. g. by the CLOUD experiment. Freshly formed particles are always added to the model at a diameter of 1.7 nm. However, adding particles to a modal distribution automatically produces particles at a lower size as part of the distribution, going down to infinitesimal small diameters. The redistribution between the different modes (more detailed outline in Pringle et al. (2010)) can then actually lead to a median diameter smaller than 1.7 nm, when larger nucleation mode particles are moved to the Aitken mode. Thus, this is only a computational artefact as a result

of the modal redistribution, while we actually do not explicitly add particles with a diameter smaller than 1.7 nm. We therefore decided to keep the definition of the UFP concentrations as it is, as otherwise (setting a lower cut to 1.7 nm) we would exclude actually nucleated particles. Additionally, we added size-resolved distributions as described in the following reply, also including the cut at 1.7 nm.

[Figure]

Figure 2: Typical particle size distribution (PSD) taken from the simulation in an urban region. The dashed lines represent the soluble and insoluble aerosol modes. The PSD is the sum of all these modes (blue line), typically dominated by the soluble modes. UFPs are defined as all particles with diameter below 100 nm (right red line), while the total particle number concentration is the full integral over the PSD. For comparison between observed and simulated concentrations, the lower bound (red dashed line) is considered as a cutoff, which depends on the measurement device in practice. The final dataset includes all UFPs without lower bound.

The size distribution in Fig. 2 in the manuscript was actually taken from a remote region over the ocean, at which the described computational artefact is most clearly apparent, which is not representative for the globe. We replaced the figure with Fig. 2 of this document taken from an urban region (Beijing).

We additionally compared the UFP concentrations with and without cut at 1.7 nm. Over continental regions, the differences are well below 10 %, with differences below 2 % in polluted regions, while over the ocean differences can increase to maximum values of 20 %. We added a short discussion on this to the "Limitations and Uncertainties" section.

10) If the authors don't want to change this, they really need to make the number concentration between 1.7nm and 100nm public so users could exclude particles that are, at face value, molecular clusters, from their analysis. In fact, I strongly encourage the authors to make public at least UFP concentrations in a couple of size ranges, irrespective of the lower cut-off. Say 50-100nm particles and 10-100nm particles. And monthly rather than annual averages. The paper will surely attract more citations that way.

We thank the reviewer for this suggestion to improve our manuscript. We added three size-resolved datasets to the published datasets (1.7-20 nm, 20-50 nm, 50-100 nm) and additionally discussed and showed these results in the manuscript and here in Fig. 3. We see a clear

[Figure]

Figure 3: Fraction of UFP concentrations in different size ranges

distinction for the three size ranges. Ocean environments are mostly dominated by freshly nucleated particles between 1.7 and 20 nm, biomass burning regions by larger particles between 50 and 100 nm, and polluted regions by Aitken mode size particles between 20 and 50 nm, that are either freshly emitted fossil fuel primary particles or grown secondary particles. As outlined in the reply to comment 2, we do not plan to publish datasets with higher temporal resolution in this scope. We focused here on the evaluation of long-term (annual averaged) concentrations. A more detailed look into seasonality will follow in future publications.

**11) The evaluation section could generally benefit from more time-resolved data. Seasonal cycles are lacking. I suggest adding another figure with seasonal cycles at many sites around the world, to complement Figure 5, which only shows sites in Leipzig.**

We refer the referee to the replies to previous comments: Studying the seasonality of UFP concentrations and size distributions is not within the scope of this manuscript and will be subject of future studies.

**12) L354 individuals (typo)**

Fixed, thanks for pointing out.

**References**

Chen, X., F. Yu, W. Yang, Y. Sun, H. Chen, W. Du, J. Zhao, Y. Wei, L. Wei, H. Du, Z. Wang, Q. Wu, J. Li, J. An, and Z. Wang (2021). "Global–regional nested simulation of particle number concentration by combing microphysical processes with an evolving organic aerosol module". In: *Atmospheric Chemistry and Physics* 21.12, pp. 9343–9366. DOI: `10.5194/acp-21-9343-2021`.

Courant, R., K. Friedrichs, and H. Lewy (1967). "On the Partial Difference Equations of Mathematical Physics". In: *IBM Journal of Research and Development* 11.2, pp. 215–234. DOI: `10.1147/rd.112.0215`.

German Climate Computing Centre (DKRZ) (2021). *Introduction to Levante.*

Jöckel, P., H. Tost, A. Pozzer, C. Brühl, J. Buchholz, L. Ganzeveld, P. Hoor, A. Kerkweg, M. G. Lawrence, R. Sander, B. Steil, G. Stiller, M. Tanarhte, D. Taraborrelli, J. van Aardenne, and J. Lelieveld (2006). "The atmospheric chemistry general circulation model ECHAM5/MESSy1: consistent simulation of ozone from the surface to the mesosphere". In: *Atmospheric Chemistry and Physics* 6.12, pp. 5067–5104. DOI: `10.5194/acp-6-5067-2006`.

Liu, M. and H. Matsui (2022). "Secondary Organic Aerosol Formation Regulates Cloud Condensation Nuclei in the Global Remote Troposphere". In: *Geophysical Research Letters* 49.18. e2022GL100543 2022GL100543, e2022GL100543. DOI: `https://doi.org/10.1029/2022GL100543`. eprint: `https://agupubs.onlinelibrary.wiley.com/doi/pdf/10.1029/2022GL100543`.

Pringle, K. J., H. Tost, S. Metzger, B. Steil, D. Giannadaki, A. Nenes, C. Fountoukis, P. Stier, E. Vignati, and J. Lelieveld (2010). "Description and evaluation of GMXe: a new aerosol submodel for global simulations (v1)". In: *Geosci. Model Dev.* 3.2, pp. 391–412. DOI: `10.5194/gmd-3-391-2010`.

Schutgens, N., S. Tsyro, E. Gryspeerdt, D. Goto, N. Weigum, M. Schulz, and P. Stier (2017). "On the spatio-temporal representativeness of observations". In: *Atmospheric Chemistry and Physics* 17.16, pp. 9761–9780. DOI: `10.5194/acp-17-9761-2017`.

Tsimpidi, A. P., V. A. Karydis, A. Pozzer, S. N. Pandis, and J. Lelieveld (2014). "ORACLE (v1.0): module to simulate the organic aerosol composition and evolution in the atmosphere". In: *Geoscientific Model Development* 7.6, pp. 3153–3172. DOI: `10.5194/gmd-7-3153-2014`.

---

## Author Comment (AC2)

**Referee comment #2:**

This study presents methodology for global modelling of surface level UFP with the ECHAM/MESSy model. Calculations are performed for 2015 and the gridded annual average values are compared to measured annual values obtained from EBAS, GAW-WDCA and ACTRIS covering stations in Europe, North America, India, China and remote regions from 2015 and other years.

The coarse resolution model results are downscaled to high spatial resolution using medium and high spatial resolution emission data together with statistical parameters from the model evaluation with measurements to give estimated annual values on a high resolution grid for the Globe.

The manuscript is a highly relevant contribution to the science revolving around UFP modelling, and it is well-written and clearly structured. I have some comments that may require a bit more work for the authors, but I leave it to the Editor to decide if major revisions are needed (I would be interested in reviewing a revised manuscript, but it is not an ultimate demand).

I encourage the authors to also make the emission data set for UFP in the 7 classes available publicly, as this would surely foster many citations.

We thank the reviewer for reviewing our manuscript and for providing helpful comments. We report the comments (grey) along with our replies (black).

Concerning the publication of the emission dataset for UFP, we need to clarify that we did not develop any new emission dataset here. Rather, we used the median diameter of the size-segregated particle number emissions from Paasonen et al. (2016) to translate primary mass emissions from CEDS for the respective years into number emissions for each sector. This is just an adaptation of the emission size distribution dataset from Paasonen et al. (2016) that is required for our model input. For future use of UFP emission data, we recommend either using the size distributions provided by Paasonen et al. (2016) or the median diameters derived by us, depending on the model requirements.

**Specific comments:**

1) In my experience, health impact studies of long-term exposure, need two major components from the air quality data: interannual variability and rural-urban contrasts (urban increment). This present study is very thorough and highly interesting for the air pollution modelling community and the research of UFP distribution and dynamics, but promoting it for health effect studies at this stage is a little premature. Often epidemiologists conducting health impact studies have very little possibility to truly realise the issues associated with data sets of this kind. I suggest to rephrase with less emphasis on the possibility for immidiate use of these data for health studies.

We thank the reviewer for this comment. We acknowledge the uncertainties and issues that still cannot be addressed in this dataset. However, we believe that the UFP dataset is of immediate use for the public health community. To substantiate this, we extended the simulation to the years 2016 and 2017 and added additional text on the (still statistically limited) interannual

[Figure]

Figure 1: Absolute interannual variation of UFP concentrations calculated based on the years 2015-2017. It is defined by the maximum minus the minimum annual averaged UFP concentrations.

variability of ultrafine particle concentration. As we provide results for health applications of long-time exposure to UFP concentrations, annual averaged UFP concentrations over 3 years will be of great importance for the community.

We present the absolute interannual variability in UFP concentrations in Fig. 1. We see a strong interannual variation, exceeding 1000 per cm$^3$ and more than 50 % along the ITCZ, that is mostly caused by meteorological differences in the different years, as 2015 was a strong El Niño year, followed by weaker La Niña and El Niño events in 2016 and 2017. This meteorological influence on UFP concentrations could be the basis for future studies. We additionally see higher interannual variations over polluted regions, although below 20 %. This is mostly due to a decreasing trend in UFP concentrations over South and East Asia, Europe and the United States. However, we see an increasing trend over parts of Africa, especially Nigeria and South Africa. We added this figure and the respective discussion to the "Results" section in the manuscript.

Regarding the urban increment, the rural-urban contrast is implicitly represented by the "high-resolution" dataset of 0.1° × 0.1°, and explicitly in our downscaling approach, where we derive a direct relation between increasing (anthropogenic) particle emissions and increasing UFP concentrations. In the future we will work on approaches to derive UFP concentrations at even higher resolution, giving way to a more detailed analysis of the rural-urban contrast.

**2) The horisontal resolution of the model is quite coarse in the simulations, and this is discussed and partly addressed in the downscaling. What about the vertical resolution/representation? A surface layer of 45-70 meters is also quite coarse for representing a measurement station at e.g. 4 meters above the surface. How is this handled? Also in connection with the down-scaling? Please share your considerations about this.**

We agree with the reviewer that the vertical extent of the surface layer adds uncertainty to the dataset. While the surface layer is situated entirely in the boundary layer most of the time, there is still a profile observed in general, as reported by Du et al. (2017), Harrison et al. (2019), Heintzenberg et al. (2011), and Villa et al. (2017), with the strongest decrease

with altitude being observed close to UFP sources. We analysed the relative change in UFP concentrations from the surface layer to the next-highest model level in Fig. 2 of this document. We find a decrease of about 20-30% in UFP concentrations on average over polluted regions, with the reduction slightly exceeding 50% over Central Africa. At higher latitudes, we even find increasing UFP concentrations at the second lowest model level.

The decrease over polluted regions is in line with the observations from Du et al. (2017), Harrison et al. (2019), and Heintzenberg et al. (2011), who observed a decrease of up to 30 % at comparable altitudes. Villa et al. (2017) even observed a decrease of up to 80%, however these measurements were performed directly at a highway and at some distance to other sources. This setup is not representative for a grid box covering approx. 180 x 180 km and tends to overestimate the effect.

Considering a linear decrease respectively increase within the range of 30% within the model surface layer would add uncertainty of ±15% (as the surface emissions will be 15% higher or lower than the grid box average concentrations) to the UFP concentrations. We added a short summary of these findings in the section on "Limitations and Uncertainties" of the manuscript.

[Figure]

Figure 2: Relative change of simulated UFP concentrations from the surface layer to the next-highest model level in %.

3) Where does equation (1) originate from? Please either include more information on the derivation or add a reference.

We thank the referee for pointing out to a more detailed description of equation (1). There is actually a typo and the (+) should have been a (·). We adjusted the equation, and added more description and a reference to Seinfeld and Pandis, 2016:

"The number of emitted aerosols ($N_{aer}$) is calculated as

$$N_{aer} = \frac{6 \cdot M_{aer}}{\pi \cdot \rho_{aer} \cdot d_{med}^3} \cdot \exp\left(-4.5 \ln^2 \sigma_{ln}\right) \tag{1}$$

where the emitted aerosol mass $M_{aer}$ is given by the respective emission dataset. $\rho_{aer}$ is the density of the considered aerosol species and $\sigma_{ln}$ is the width of the log-normal mode in the model. The fraction is just the geometrical derivation of the number of spherical particles from total

mass, given a diameter $d_{med}$, while the exponential function corrects for the lognormal volume distribution (compare equations (8.34) and (8.51) from Seinfeld and Pandis, 2016)."

**4) In the methods description, references for the methodology for median emission diameter for SLV is missing.**

There are no aerosol emissions for the sector SLV, but only gas phase emissions. Thus, no emission diameter is needed. We added this information in the description of the sectors.

**5) I realise the global model is very complex and time consuming to run, but for the argumentation that it cannot be run for more than 1 year to stand more strongly, please add some numbers on complexity, e.g. number of chemical components included, how long does a simulation take on a state-of-the-art server etc.**

As mentioned in comment 1, we extended the simulation up to 2017, covering 3 years of analysis. A typical simulation of 1 year takes roughly 2 days on 3 *AMD 7763 CPU* nodes with 128 cores each at the German Climate Computing Centre (DKRZ, 2021), and thus 8 days for the total simulation run (including spin-up). Our chemical mechanism comprises more than 100 gas phase species and more than 250 reactions (Jöckel et al., 2006). Additionally, we include aerosol microphysics and explicit aerosol-gas equilibrium calculations (Pringle et al., 2010; Tsimpidi et al., 2014). We also added this additional information in the revised manuscript.

**6) Line 266: there have been made attempts to model street-scale in individual countries/cities using other approaches, see e.g. Ketzel et al., 2021 (this could maybe also be relevant to include in the discussion)**

Thank you for the suggestion. Firstly, we used the reference for a short comparison to our "Results": "Ketzel et al. (2021) simulated PNCs at street- and address-level in the same order of magnitude as our simulation up to urban background level, but with much higher peak values reaching more than $30,000$ per $cm^3$ in traffic hotspots, which we cannot resolve.". Secondly, we also extended the "Limitations and Uncertainties" section based on the same citation: "A next step could be to apply high-resolution dynamical downscaling of concentrations guided by comprehensive measurements of which data may become available in the future, either using machine learning methods or a combination of operational street pollution models and human exposure modelling as done by Ketzel et al. (2021).".

**7) The authors discuss the issues related to comparing model results for 2015 with measurements from other years. It is an understandable approach, given the small number of measurements available, but it seems problematic when also the measurement-model relations from non-matching years are used to guide the downscaling of results for 2015 using emissions from 2015 (are the emissions from 2015?). Please either repeat the derivation of the linear fit in Figure 7 with only 2015 matches, or discuss the potential issues in depth.**

We understand the concerns of the reviewer to include measurements of varying years to scale down UFP concentrations based on 2015 emissions. However, there is only one 2015 measurement from India and one from China. Combined with the request of the comment below of subdividing the observations into a training and validation dataset, this would not leave us with sufficient statistics. We would like to to emphasize that no complex model is trained using the observational data. Rather, the fit parameter $c$ is just an estimate for the order of magnitude, by which the UFP concentrations are influenced by local emissions. This fit parameter is

optimised for the available observations.

It would be desirable to do the downscaling and evaluation based on only one year, but we would need more measurements for that. As pointed out by the reviewer, we included a discussion on the potential issues of this methodology in the limitations and uncertainties section:

"The downscaling was performed using evaluation results from all years, while the simulation results were only taken from 2015. Thus, there are potential interfering effects from the downscaling and general temporal trends as we can see from the study of interannual variability in Sect. 4."

**8) Using measurements to guide downscaling and then the same measurements to validate the downscaled model results can be considered problematic. Again I understrand the approach, as not many measurements are there to use, but would it be possible to use one set of data to do carry out the downscaling, and then evaluate with another separate set (like it is done in data assimilation)?**

We thank the referee for the suggestion. However, as mentioned in the previous comment, the observations are only used to estimate the order of magnitude of the influence of local anthropogenic emissions on UFP concentrations. For this purpose, we wanted to exploit the maximum number of the limited observations to get the best estimate.

As we are in favor of an independent validation as well, we added an analysis, randomly subdividing the observations into a training (25 stations) and validation (24 stations) dataset, performed the analysis 5000 times and show the improvement gained from this independent analysis. The results for the improvement in the validation dataset are shown in the revised manuscript and here (Fig. 3), supporting the claim that the parameter estimation is robust and largely independent from the chosen stations.

The fit parameters $c_{\mathrm{CEDS}}$ and $c_{\mathrm{EDGAR}}$ range between 0.07 and 0.23 in 90% of the runs, while the average values are very close to the values derived from the complete dataset, and all derived fit parameters are greater than 0. The RMSLE consistently decreases with rising order of downscaling, with an average improvement of 0.11 in total (95.6% of the time RMSLE decreases), while the logarithmic correlation increases on average by 0.07 (improves 95.1% of the time). The bias $\overline{\mathrm{M/O}}$ evolves from 0.78 on average in the model output resolution (2.4% of the time between 0.9 and 1.1) to 0.92 after the CEDS downscaling (55.3% between 0.9 and 1.1) to 1.04 (66.2% between 0.9 and 1.1) after the final downscaling, slightly overestimating the observed values. This analysis shows that the fit parameters can be transferred to stations that have not been seen by the training dataset. However, as we want to maximise the number of measurement stations to use for the global downscaling, we use the complete evaluation results for the downscaling as described before.

**9) Introduction: One of the issues with UFP measurements is the lack of consolidated guidelines for methodology. Could be good to mention this issue in the introduction, and perhaps also discuss at a later stage.**

We added a short discussion on this topic in the introduction and "Limitations and Uncertainties" section, based on work of Trechera et al. (2023): "Furthermore, there are no clear methodology guidelines for measuring PSDs or particle number concentrations (PNCs) (Trechera et al., 2023) with new recommendations developing recently (ACTRIS, 2021; CEN/TS 16976:2016, 2016; CEN/TS 17434:2020, 2020), and measurement size limits vary greatly."

[Figure]

Figure 3: Results of 5000 independent training and validation runs for the fit parameter estimation (25 stations in training dataset, 24 in validation dataset). The fit parameters are displayed in the top panel, while the statistical measures are displayed below at different orders of downscaling, along with the absolute change of RMSLE and the logarithmic correlation. We show box-whisker plots, where the boxes illustrate the 25-75% interval and the whiskers the 5-95% interval. Median values are marked by the vertical white line and mean values by the light-grey circles.

**10) Examples of other modelling studies for Europe that could be mentioned (to refer to multi-year modelling studies) are**

- **Fountoukis, C., Riipinen, I., Denier van der Gon, H.A.C. et al, 2012. Atmos. Chem. Phys. 12, 8663–8677.**

- **Kukkonen, J., Karl, M., Keuken, M.P., et al 2016. Model Dev. (GMD) 9, 451–478.**

- **Frohn, L.M., Ketzel, M., Christensen, J.H. et al.,, 2021. Atm. Env. Vol. 264, 118631.**

We thank the reviewer for pointing out this additional literature. We refer to all these studies in the introduction, and briefly discuss differences with the results from Fountoukis et al. (2012) in the "Results" section: "Our simulated UFP concentrations are much lower than the modelled results from Fountoukis et al. (2012), that reach around $20,000$ per $cm^3$, even for background regions in Europe in May 2008, and up to more than $100,000$ per $cm^3$ in local hotspots in

Southeastern Europe. However, the general increasing trends from Western to Eastern Europe agree."

**11) Line 328: how did you estimate surface level concentrations from a grid cell? Is this methodology also used for the other regions (than remote)?**

We apologize for formulating this in a misleading way. We did not estimate surface level concentrations from the grid cell. Surface level referred to the lowest vertical model level and "the corresponding grid cell" just referred to the horizontal extent. We reformulated to: "... and compared them to $UFP_M$ concentrations in the lowest vertical model level of the corresponding horizontal grid cell at the respective day of the year."

**Technical comments:**

**12) Last sentence of the introduction – how does population density introduce inconsistencies? Surely it is the model representation that introduce inconsistensies?**

Yes, this was improperly worded. We reformulated to: "... introduced by the inability of the coarse model resolution to capture population density and air pollution gradients"

**13) What is shown in Figure 3: lowest model layer average, or surface layer (and at what height?)**

We show the average of the lowest model level. The height of the surface level is varying between 45-70m depending on latitude and season, as described in lines 80-81. We added the following in the caption of Figure 3: "... and from the lowest vertical model level (surface level)."

**14) Figure 5: the daily fluctuating concentrations are difficult to see in the figure, please consider replotting**

[Figure]

Figure 4: Monthly averaged UFP concentrations in the simulation (grey, dashed) and measured at different stations in Leipzig, Germany (thick solid lines). The daily fluctuating UFP concentrations are shown with thin, transparent lines in the same color.

We thank the reviewer for pointing this out. We made a new figure with some adjustments and a slightly modified color scheme (see Fig. 4).

**15) Would be good to make the text consistent across tables, e.g. table 2 should use the formulation from table 4 considering modelling results from 2015 and measurements from other years.**

Added in table 2, thank you for pointing out.

**16) I am assuming that emission data are from 2015, but please specify the year of the emission data used.**

Yes, the emission data was used for the respective simulated years. We added the years for the description of the nudging data, and subsequently referred to the emissions datasets for the "respective years".

**17) Table 3: a typo, $PNC_M$ and $PNC_M$ should be $PNC_O$ and $PNC_M$.**

Fixed, thank you for pointing out.

**18) Line 283: what is meant?**

We are unsure what the referee is referring to, and assume it is the phrase "days with valid measurements". The observations contained days, where no measurements were specified, i.e. missing values were introduced. We excluded these days in the analysis.

**19) Line 296: is there really only one grid cell covering Delhi? Or should it be grid cells**

Yes, there is only one model grid cell covering Delhi and New Delhi, as can also be seen in Figure 8 on the downscaling method (the model grid box has a horizontal resolution of approximately $180 \times 180$ km). In the downscaled version, several grid cells represent Delhi, as apparent by the differing values of $UFP_R$ in Table 4, and pointed out in lines 417-419.

**20) Was the population data regridded to the MESSy grid? Or how did you determine the population density for each of the points in Figure 6**

We used the population density dataset at a resolution of $0.1° \times 0.1°$ and evaluated it at the coordinates of the measurement station. Thus, the values of population density are taken from a dataset with finer resolution than the simulation. We did this to illustrate that there is a model underestimation at high local population density (urban regions) and overestimation at low local population density, i. e. $UFP_M$ is dsrongly diluted by the surroundings, while the population density data is only to a lesser extent.

**21) Line 335: "with a higher than model resolution" – what is meant?**

We reformulated to: "sampled at the measurement stations with a resolution of $0.1°\times0.1°$"

**22) Line 352: indivuduals should be individuals**

Fixed, thank you for pointing out.

**23) Line 378: "referred to as $PAE_{GB}$"**

We reformulated as $PAE_{GB}$ was not used in the following anyways: "and the relation to the respective grid box average at coarser resolution"

**24) Figure 9: the symbols for "Before DS" are tricky to see in the plot, please consider using another font or color.**

[Figure]

Figure 5: Improved Fig. 9

We thank the reviewer for pointing this out. We made a new figure with a slightly modified color scheme (see Fig. 5).

**References**

ACTRIS (2021). *ACTRIS Recommendation for mobility particle size spectrometer measurements.*

CEN/TC 264/WG 32 - Air quality - Determination of the particle number concentration (2016). *CEN/TS 16976:2016: Ambient air - Determination of the particle number concentration of atmospheric aerosol.*

CEN/TC 264/WG 32 - Air quality - Determination of the particle number concentration (2020). *CEN/TS 17434:2020: Ambient air - Determination of the particle number size distribution of atmospheric aerosol using a Mobility Particle Size Spectrometer (MPSS).*

Du, W., J. Zhao, Y. Wang, Y. Zhang, Q. Wang, W. Xu, C. Chen, T. Han, F. Zhang, Z. Li, P. Fu, J. Li, Z. Wang, and Y. Sun (2017). "Simultaneous measurements of particle number size distributions at ground level and 260 m on a meteorological tower in urban Beijing, China". In: *Atmospheric Chemistry and Physics* 17.11, pp. 6797–6811. DOI: 10.5194/acp-17-6797-2017.

Fountoukis, C., I. Riipinen, H. A. C. Denier van der Gon, P. E. Charalampidis, C. Pilinis, A. Wiedensohler, C. O'Dowd, J. P. Putaud, M. Moerman, and S. N. Pandis (2012). "Simulating ultrafine particle formation in Europe using a regional CTM: contribution of primary emissions versus secondary formation to aerosol number concentrations". In: *Atmospheric Chemistry and Physics* 12.18, pp. 8663–8677. DOI: 10.5194/acp-12-8663-2012.

German Climate Computing Centre (DKRZ) (2021). *Introduction to Levante.*

Harrison, R. M., D. C. S. Beddows, M. S. Alam, A. Singh, J. Brean, R. Xu, S. Kotthaus, and S. Grimmond (2019). "Interpretation of particle number size distributions measured across an urban area during the FASTER campaign". In: *Atmospheric Chemistry and Physics* 19.1, pp. 39–55. DOI: 10.5194/acp-19-39-2019.

Heintzenberg, J., W. Birmili, R. Otto, M. O. Andreae, J.-C. Mayer, X. Chi, and A. Panov (2011). "Aerosol particle number size distributions and particulate light absorption at the ZOTTO tall tower (Siberia), 2006–2009". In: *Atmospheric Chemistry and Physics* 11.16, pp. 8703–8719. DOI: 10.5194/acp-11-8703-2011.

Jöckel, P., H. Tost, A. Pozzer, C. Brühl, J. Buchholz, L. Ganzeveld, P. Hoor, A. Kerkweg, M. G. Lawrence, R. Sander, B. Steil, G. Stiller, M. Tanarhte, D. Taraborrelli, J. van Aardenne, and J. Lelieveld (2006). "The atmospheric chemistry general circulation model ECHAM5/MESSy1: consistent simulation of ozone from the surface to the mesosphere". In: *Atmospheric Chemistry and Physics* 6.12, pp. 5067–5104. DOI: 10.5194/acp-6-5067-2006.

Ketzel, M., L. M. Frohn, J. H. Christensen, J. Brandt, A. Massling, C. Andersen, U. Im, S. S. Jensen, J. Khan, O.-K. Nielsen, M. S. Plejdrup, A. Manders, H. D. van der Gon, P. Kumar, and O. Raaschou-Nielsen (2021). "Modelling ultrafine particle number concentrations at address resolution in Denmark from 1979 to 2018 - Part 2: Local and street scale modelling and evaluation". In: *Atmospheric Environment* 264, p. 118633. ISSN: 1352-2310. DOI: https://doi.org/10.1016/j.atmosenv.2021.118633.

Paasonen, P., K. Kupiainen, Z. Klimont, A. Visschedijk, H. A. C. Denier van der Gon, and M. Amann (2016). "Continental anthropogenic primary particle number emissions". In: *Atmospheric Chemistry and Physics* 16.11, pp. 6823–6840. DOI: 10.5194/acp-16-6823-2016.

Pringle, K. J., H. Tost, S. Metzger, B. Steil, D. Giannadaki, A. Nenes, C. Fountoukis, P. Stier, E. Vignati, and J. Lelieveld (2010). "Description and evaluation of GMXe: a new aerosol submodel for global simulations (v1)". In: *Geosci. Model Dev.* 3.2, pp. 391–412. DOI: 10.5194/gmd-3-391-2010.

Seinfeld, J. H. and S. N. Pandis (2016). *Atmospheric chemistry and physics: from air pollution to climate change.* John Wiley & Sons.

Trechera, P., M. Garcia-Marlès, X. Liu, C. Reche, N. Pérez, M. Savadkoohi, D. Beddows, I. Salma, M. Vörösmarty, A. Casans, J. A. Casquero-Vera, C. Hueglin, N. Marchand, B. Chazeau, G. Gille, P. Kalkavouras, N. Mihalopoulos, J. Ondracek, N. Zikova, J. V. Niemi, H. E. Manninen, D. C. Green, A. H. Tremper, M. Norman, S. Vratolis, K. Eleftheriadis, F. J. Gómez-Moreno, E. Alonso-Blanco, H. Gerwig, A. Wiedensohler, K. Weinhold, M. Merkel, S. Bastian, J.-E. Petit, O. Favez, S. Crumeyrolle, N. Ferlay, S. Martins Dos Santos, J.-P. Putaud, H. Timonen, J. Lampilahti, C. Asbach, C. Wolf, H. Kaminski, H. Altug, B. Hoffmann, D. Q. Rich, M. Pandolfi, R. M. Harrison, P. K. Hopke, T. Petäjä, A. Alastuey, and X. Querol (2023). "Phenomenology of ultrafine particle concentrations and size distribution across urban Europe". In: *Environment International* 172, p. 107744. ISSN: 0160-4120. DOI: https://doi.org/10.1016/j.envint.2023.107744.

Tsimpidi, A. P., V. A. Karydis, A. Pozzer, S. N. Pandis, and J. Lelieveld (2014). "ORACLE (v1.0): module to simulate the organic aerosol composition and evolution in the atmosphere". In: *Geoscientific Model Development* 7.6, pp. 3153–3172. DOI: 10.5194/gmd-7-3153-2014.

Villa, T., E. Jayaratne, L. Gonzalez, and L. Morawska (2017). "Determination of the vertical profile of particle number concentration adjacent to a motorway using an unmanned aerial

vehicle". In: *Environmental Pollution* 230, pp. 134–142. ISSN: 0269-7491. DOI: `https://doi.org/10.1016/j.envpol.2017.06.033`.

---

## Author Response (AR2)

**Referee comment**

The authors present a nice analysis and dataset of ultrafine particle concentrations using a state-of-the-art model. They have mostly taken my previous comments on board, and the paper is almost ready.

Two key reasons why I still have "minor comments" and not just "technical corrections".

We thank the reviewer for the second round of reviewing our manuscript and for providing helpful comments. We report the comments (grey, bold) along with our replies (black).

**1) First, I encourage the authors to discuss spatial as well as temporal representativeness uncertainties (as per Schutgens et al) in more detail as they are so critical to the model-observation comparisons in urban regions. This valuable prior work sets the context for much of the authors' study and is only acknowledged in a cursory comment in the discussion of uncertainties. A measurement at a single location within a large model gridbox containing exactly the spatial gradients in aerosol concentrations that the authors address with their downscaling approach is unlikely to represent the average concentration in that grid cell.**

We thank the reviewer for the emphasis on the representativeness. In the first round of replies, the comment was understood to be mainly focused on the comparison to the ATom flights, as we cannot address the temporal representation error here and Schutgens et al. (2017) does not target in-situ ground measurements in great detail. However, after the more detailed study of Schutgens et al. (2016a), we agree that especially the spatial representation error is a perfect setting for the introduction for the downscaling. In fact, we try to address and reduce spatial representative errors by the downscaling without explicitly mentioning it in the manuscript.

In general, we believe that we have three different types of representation errors:

1. Purely spatial representation errors only due to the extent of the grid box at stations, for which timeline measurements of 2015 (mostly hourly) were available and we collocated our simulation with the observations according to Schutgens et al. (2016b), i. e. all observations from the EBAS database.

2. Spatio-temporal representation errors for measurement stations, for which only annual averages were available or the years were differing from the evaluation year (2015).

3. (Spatio-)temporal representation errors for the aircraft measurements from ATom, for which we used daily averaged model output and the measurement years are differing from the simulation year. The spatial representation error is reduced by the fact, that we have several measurements for each grid box.

The horizontal downscaling of UFP concentrations addresses and reduces the spatial representation error for types 1 and 2, while the temporal representation error of types 2 and 3 cannot be addressed.

We introduced and applied the representation error concept from Schutgens in more detail in the revised manuscript.

2) Second, thanks to the authors' helpful responses to my comments, it is now more obvious that their approach to comparing model to measurements is not quite correct, because they do not explicitly exclude particles which were too small to be seen by the counters used to make the measurements. Usually, the people who took the measurements are forthcoming with information about their lower size cut-offs, and most similar studies I am aware of do explicitly consider them (e.g. Ketzel et al 2021 as referenced in the paper, or Spracklen et al, ACP 2010: https://acp.copernicus.org/articles/10/4775/2010/). I think the authors need to be clearer about the short-cut they took here in the paper text, but I personally would not try to insist that the authors do the comparison more correctly with the lower size cut-offs included in their analysis, because they do present an analysis of the uncertainty introduced by not doing this.

We believe that we were not clear enough in our previous replies. In fact, in contrast to the referee comment, we do explicitly exclude particles which were too small to be seen by the measuring instruments in the model-observation comparisons. In Eq. (2) and the subsequent lines we outline how we calculate UFP numbers based on $D_{up}$ and $D_{low}$:

"... where $D_{up}$ is the fixed upper bound of 100 nm and $D_{low}$ is the variable lower bound associated with the measurement device. For the final dataset we report the total number of UFPs, and thus $D_{low}$ is set to 0 nm and the second error function in Eq. (2) evaluates to $-1$."

Thus, we set $D_{low}$ to the lower cutoff diameter of the instrument ("Cut" in Tables 1, 2, 4 and 5) in the observation-simulation comparison. The only exception is the comparison to PNC measurements from the EBAS database, as no cutoff diameters were available (Table 3 and last three measurements presented in Table 6). We stated this in the caption of Table 3: "There is no particle size cutoff value given in the datasets, and thus none is applied on the simulation."

The exact procedure was also described in Section 4.1, lines 275-277: "The daily averaged number concentrations of the model aerosol modes were integrated for the same size region (from the lower measurement cut up to the highest measurement bin with a mean diameter below 100 nm) according to Eq. (2)."

The procedure how we calculate the resulting global dataset of UFP concentrations (setting $D_{low} = 0$ as outlined in detail in the replies to the first round of comments) should not be mistaken with the procedure we perform for the evaluation of the model results using observations (setting $D_{low}$ to the lower cutoff of the measurement device - "Cut" in the evaluation tables).

We included the information additionally in the caption of Tables 1, 4 and 5, while it was already provided in Table 2. We removed Fig. 6 of the revised manuscript as we believe that it is not of great importance for the paper and can lead to misunderstandings, i. e. in Fig. 6 we present UFP concentrations of the simulation with $D_{low} = 0$ (in contrast to the actual evaluation presented in the table) along with measurements with differing $D_{low}$. Additionally, we slightly reformulated the description of our evaluation procedure to make it clearer.

**Minor comments:**

3) L60 methodology → methodological

Fixed, thank you for pointing out.

4) L258 "increased" compared to what? You probably mean "significant"?

We rephrased this to: "We additionally find absolute interannual variation exceeding 1000 per cm$^3$ over polluted regions, although below 20 % in relative terms."

5) "respectively" → "or"

Adjusted, thank you for pointing out.

6) "Additional uncertainties are introduced by the missing guidelines for PSD and PNC measurements along with different measurement size ranges (Trechera et al., 2023)." —this sentence is unclear, and needs to be rephrased to specify what these "guidelines" are. Also it's not obvious why different measurement size ranges introduce uncertainty, unless the comparison is done without matching the size range between measurements and simulations: in line with my comment above, I think the authors need to specify more explicitly that (if I understand correctly) they made no attempt to do this.

The referee is right, that the formulation is unclear. It should actually be a reference on missing guidelines in the measurements, that we shortly discussed in the introduction. In fact, it does not add uncertainties to the simulation and evaluation, especially because we are using the same size range for the simulation as in the measurements for the evaluation (compare reply on comment (2)). Therefore, we removed this sentence.

7) L533 the "free ocean" is not well-defined, at least not to me, and I think representativeness uncertainties are more relevant to the urban regions the authors mainly focus on, where there are strong urban -rural contrasts and intra-city inhomogenenities. See earlier comment.

Compare reply on comment (1): We removed this part and reworked the manuscript with respect to the inclusion of the representation error concept based on the work of Schutgens et al. (2016a,b, 2017).

**References**

Schutgens, N. A. J., E. Gryspeerdt, N. Weigum, S. Tsyro, D. Goto, M. Schulz, and P. Stier (2016a). "Will a perfect model agree with perfect observations? The impact of spatial sampling". In: *Atmospheric Chemistry and Physics* 16.10, pp. 6335–6353. DOI: 10.5194/acp-16-6335-2016.

Schutgens, N. A. J., D. G. Partridge, and P. Stier (2016b). "The importance of temporal collocation for the evaluation of aerosol models with observations". In: *Atmospheric Chemistry and Physics* 16.2, pp. 1065–1079. DOI: 10.5194/acp-16-1065-2016.

Schutgens, N. A. J., S. Tsyro, E. Gryspeerdt, D. Goto, N. Weigum, M. Schulz, and P. Stier (2017). "On the spatio-temporal representativeness of observations". In: *Atmospheric Chemistry and Physics* 17.16, pp. 9761–9780. DOI: 10.5194/acp-17-9761-2017.